# BroGNet: Momentum Conserving Graph Neural Stochastic Differential Equation for Learning Brownian Dynamics

**Suresh Bishnoi, Jayadeva, Sayan Ranu, N. M. Anoop Krishnan** *
Indian Institute of Technology Delhi, Hauz Khas, New Delhi, India 110016
`{srz208500,jayadeva,sayanranu,krishnan}@iitd.ac.in`

## Abstract

Neural networks (NNs) that exploit strong inductive biases based on physical laws and symmetries have shown remarkable success in learning the dynamics of physical systems directly from their trajectory. However, these works focus only on the systems that follow deterministic dynamics, such as Newtonian or Hamiltonian. Here, we propose a framework, namely *Brownian graph neural networks* (BroGNet), combining *stochastic differential equations (SDEs)* and GNNs to learn Brownian dynamics directly from the trajectory. We modify the architecture of BroGNet to enforce linear momentum conservation of the system, which, in turn, provides superior performance on learning dynamics as revealed empirically. We demonstrate this approach on several systems, namely, linear spring, linear spring with binary particle types, and non-linear spring systems, all following Brownian dynamics at finite temperatures. We show that BroGNet significantly outperforms proposed baselines across all the benchmarked Brownian systems. In addition, we demonstrate zero-shot generalizability of BroGNet to simulate unseen system sizes that are two orders of magnitude larger and to different temperatures than those used during training. Finally, we show that BroGNet conserves the momentum of the system resulting in superior performance and data efficiency. Altogether, our study contributes to advancing the understanding of the intricate dynamics of Brownian motion and demonstrates the effectiveness of graph neural networks in modeling such complex systems.

## 1 Introduction and Related Works

Learning the dynamics of physical systems directly from their trajectory is an active area of research due to their potential applications in materials modeling Park et al. (2021), drug discovery Vamath-evan et al. (2019), motion planning Ni & Qureshi (2022), robotics Sanchez-Gonzalez et al. (2019); Greydanus et al. (2019), and even astrophysics Sanchez-Gonzalez et al. (2020). Recent works demonstrated that physics-based inductive biases could enable the learned models to follow conservation laws such as energy and momentum while also simplifying the learning making the model data efficient Thangamuthu et al. (2022); Finzi et al. (2020). Among these, a family of models, such as Lagrangian or Hamiltonian neural networks Bhattoo et al. (2023; 2022); Lee et al. (2021); Sanchez-Gonzalez et al. (2019); Greydanus et al. (2019) and Neural ODEs Bishnoi et al. (2022); Zhong et al. (2019); Gruver et al. (2021); Chen et al. (2018), enforces the physics-based inductive biases in a strong sense. Here, a governing ordinary differential equation (ODE) is used along with a neural network to learn the abstract quantities, such as energy or force, directly from the trajectory of the system. These models have shown remarkable success in learning the dynamics of a variety of systems in an efficient fashion, *viz.*, particle-based systems Bishnoi et al. (2022); Thangamuthu et al. (2022), atomistic dynamics Park et al. (2021); Huang et al. (2022), physical systems Bhattoo et al. (2023); Sanchez-Gonzalez et al. (2019); Greydanus et al. (2019), and articulated systems Bhattoo et al. (2022).

Despite their success, these works focus on purely deterministic systems where the dynamics are governed by an ODE Karniadakis et al. (2021); Thangamuthu et al. (2022); Zhong et al. (2021). An alternative approach to model physical systems is to formulate the governing equation as a *stochastic*

---

*SB: School of Interdisciplinary Research, J: Department of Electrical Engineering, SR: Department of Computer Science, NMAK: Department of Civil Engineering, SR and NMAK: Yardi School of Artificial Intelligence (joint appointment).

differential equation (SDE), for instance, *Brownian* dynamics (BD)—widely used to study the dynamics of particles in a fluid or solvent Van Gunsteren & Berendsen (1982). Brownian dynamics simulations have proven to be highly valuable in numerous scientific disciplines. In physics, they have been used to study the diffusion of particles Chung et al. (1999), the self-assembly of colloidal systems Noguchi & Takasu (2001), and the behavior of polymers in solution Helfand et al. (1980). In chemistry, Brownian dynamics simulations have shed light on reaction kinetics Northrup & Erickson (1992), molecular diffusion Madura et al. (1995), and the behavior of macromolecules Cates (1985). In biology, they have provided insights into the movement of cells Klank et al. (2018), the folding of proteins Karplus & Weaver (1994), and the dynamics of biomolecular complexes Gabdoulline & Wade (2001). Despite their importance in such wide domains, while there have been approaches to inferring the nature of Brownian dynamics from observations in a statistical sense Gnesotto et al. (2020); Genkin et al. (2021), to the best of the authors' knowledge, no attempt has been made to learn the dynamics of Brownian systems in particular, or SDEs in general, from their trajectory.

An alternate approach involves learning the dynamics directly using the graph representation of the data Poli et al. (2021); Chamberlain et al. (2021); Eliasof et al. (2021); Gravina et al. (2023); Rusch et al. (2022); Wang et al. (2019); Eliasof et al. (2023); Wu et al. (2023). Here, a physical system is modeled using a graph neural network (GNN) where the nodes represent the particles and the edges represent the interactions. To these models, additional inductive bias in terms of physics-informed loss functions ("soft constraints") or architectures ("hard constraints") can be employed. Diverse approaches integrating soft and hard constraints have advanced the understanding and learning of dynamical systems. Previous works explored the utilization of coupled ODEs for real-time diagnosis of COVID-19 spread Dandekar et al. (2020), capturing system invariants Zhang et al. (2023), and incorporating conservation laws Hansen et al. (2024), to name a few. Furthermore, the significance of physics-informed inductive biases to enhance the performance of GNNs has also been studied Thangamuthu et al. (2022). Architectural modifications for hard constraints include graph neural ODEs, and Lagrangian and Hamiltonian graph neural networks Poli et al. (2021); Sanchez-Gonzalez et al. (2019); Bhattoo et al. (2023; 2022). Haber & Ruthotto (2017) presents an architecture that mitigates gradient instability with ODEs and Hamiltonian-inspired forward propagation. E et al. (2017) successfully models complex dynamics through a connection with backward stochastic differential equations. However, most of these works focus on deterministic systems, and there are limited efforts to extend them to stochastic systems Eliasof et al. (2023).

Here, we propose BROGNET, which can learn the Brownian dynamics of a system directly from the trajectory. Exploiting a GNN-based framework, we show that the Brownian dynamics can be learned from small systems, which can then be generalized to arbitrarily large systems. The major contributions of the work are as follows.

- **Learning stochastic dynamics from trajectory:** We present a framework, BROGNET, that learns the abstract quantities governing the dynamics of stochastic systems directly from their trajectory. While demonstrated for Brownian dynamics, the framework presented is generic and may be applied to other SDEs as well.
- **Graph-based modeling for zero-shot generalizability:** We model the physical system as a graph with nodes representing particles and edges representing their interactions, which can then be used to learn BD. The inductive graph-based modeling naturally imparts the desirable properties of *zero-shot generalizability* to unseen system sizes and *permutation invariance*. In addition, we also demonstrate generalizability to unseen temperatures in a zero-shot fashion.
- **Momentum conservation:** We demonstrate that the proposed framework follows Newton's third law, thereby conserving the total linear momentum of the system. This feature, in turn, provides superior performance for the model.

## 2 BACKGROUND ON BROWNIAN DYNAMICS

The fundamental concept behind Brownian dynamics is the influence of thermal fluctuations on the motion of particles. In a fluid, particles are subject to random collisions with the surrounding molecules, which lead to erratic motion characterized by continuous changes in direction and speed (see the videos in the supplementary for a visual depiction). The behavior of particles undergoing Brownian motion can be described mathematically using SDEs, such as over-damped *Langevin* equations Coffey & Kalmykov (2020). These equations incorporate deterministic forces, such as external potentials or interactions between particles, as well as stochastic forces that represent the

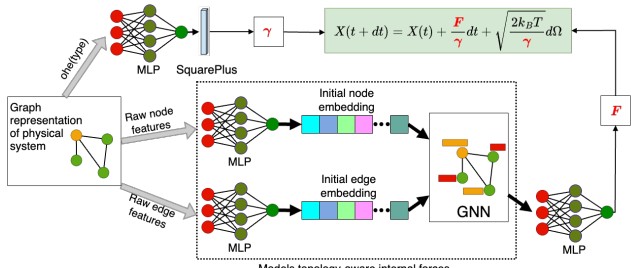

Figure 1: BROGNET's architecture. MLP refers to multilayer perception, GNN to graph neural networks and ohe to one-hot encoding.

random effects of the surrounding fluid. By numerically integrating these equations, it is possible to simulate and analyze the dynamic behavior of the particles over time.

Consider a system of $N$ particles with masses $M$ and interacting with each other through an interaction potential $U(X_t)$, where $X_t$ represents the position vectors of all the particles at time $t$, constituting a time-dependent random variable. The Langevin equation governing the dynamics of this system is given by Coffey & Kalmykov (2020)

$$M\ddot{X}_t = -\nabla U(X_t) - \zeta M \dot{X}_t + \sqrt{2M\zeta k_B T}\,\Omega_t \tag{1}$$

where $-\nabla U(X_t) = F(X_t)$ is the force due to the inter-particle interactions, $\zeta$ is the damping constant, $T$ is the temperature, $k_B$ is the Boltzmann constant, and $\Omega_t$ is a delta-correlated stationary Gaussian process with zero mean that follows $\langle \Omega_t \rangle = 0$ and $\langle \Omega_t . \Omega_{t'} \rangle = \delta(t - t')$, where $\delta$ is the standard Dirac-delta function. In the over-damped limit of Langevin dynamics, where no acceleration exists, the equation reduces to Brownian dynamics as Brańka & Heyes (1998)

$$\frac{dX_t}{dt} = \underbrace{\frac{-\nabla U(X_t))}{\gamma}}_{\text{drift term}} + \underbrace{\frac{\sqrt{2\gamma k_B T}}{\gamma}}_{\text{diffusion term}}\,\Omega_t \tag{2}$$

where $\gamma = M\zeta$. Thus, the evolution of a system can be obtained by numerically integrating the SDE given by Eq. 2. The first term in the RHS of Eq. 2 is the deterministic inter-particle interaction (drift term), and the stochastic nature in the dynamics comes from the second term (diffusion).

## 3 BROGNET: A GRAPH NEURAL SDE APPROACH

To learn the trajectories of a multi-particle system governed by Brownian dynamics, we design a neural network called BROGNET. BROGNET is essentially a Graph Neural SDE—inspired by Graph Neural ODE, which parametrizes an ODE using a GNN. Thus, BROGNET parameterizes the stochastic dynamics $B(X_t)$ using a neural network and learns the approximate function $\hat{B}(X_t)$ by minimizing the loss between the *distribution* of ground truth trajectories with that of the predicted trajectories. However, in contrast to classical dynamics, the loss is applied over a distribution of positions rather than the exact deterministic trajectory due to the stochastic nature of the dynamics. We discuss the exact loss function in detail later.

The learning process is, therefore, decomposed into two factors: the *standard deviation* associated with the ground-truth stochastic process corresponding to each particle and their *expected positions*. In BROGNET, the standard deviation is learned through an MLP. On the other hand, the expected positions are learned through a Graph Neural SDE. The choice of using a GNN to model interaction dynamics is motivated by two key observations. First, a GNN allows us to be inductive, where the number of model parameters is independent of the number of particles in the system. Hence, it enables zero-shot generalizability to systems of arbitrary unseen sizes. In addition, a GNN allows *permutation-invariant* modeling of the system. Below, we discuss the details.

### 3.1 THE GNN ARCHITECTURE

BROGNET models the interaction dynamics among participating particles as a graph. The graph topology is used to learn the approximate dynamics $\hat{B}$ by minimizing the distance between distributions representing the true and predicted positions. Note that henceforth all the variables with a hat () correspond to the approximate variable learned by the machine learning model.

**Graph structure.** We represent an $N$-particle system as an directed graph $\mathcal{G} = \{\mathcal{V}, \mathcal{E}\}$. $\mathcal{V}$ represents the set of nodes, where each node represents a particle. Similarly, $\mathcal{E}$ represents interactions or

connections among particles in the form of edges. For example, in a ball-spring system or colloidal gels, the balls or colloid particles correspond to nodes, and the springs or the interactions between the colloidal particles correspond to edges. Note that in the case of systems such as springs, where explicit connections are pre-defined, the graph structure remains static. However, in the case of systems like colloidal gels, the edges are defined based on a neighborhood cut-off, and hence the graph structure can be dynamic, with the edges between particles changing as a function of the particle configuration. As we will see in the next paragraph, a directed graph is required only because the edge weights we use are directional.

**Input features.** Each node corresponds to a particle $i$ and is defined by its intrinsic features, such as the particle's type ($\tau_i$), position ($X_{i,t}$), and velocity ($\dot{X}_{i,t}$) at time $t$. The position and velocities are 3-dimensional tuples corresponding to the $x$, $y$ and $z$ dimensions. The particle type distinguishes between different characteristics, such as varying masses or friction coefficient for balls or particles. For brevity, $X_{i,t}$ is referred to as $X_i$ henceforth unless specified otherwise. On the other hand, each edge represents the connection between nodes and is represented by the 3-dimensional edge feature $w_{ij} = (X_i - X_j)$, which signifies the relative displacement of the connected nodes along each coordinate. Note that the edge weights are asymmetric due to the vectorial nature of the input.

**Summary of neural architecture.** Fig. 1 provides a pictorial description of BROGNET. In the pre-processing layer, we utilize Multilayer Perceptrons (MLPs) to create dense vector representations for each node $v_i$ and edge $e_{ij}$. In systems where internal forces play a significant role in governing dynamics, the structure's topology holds crucial importance. To capture this dependency, we employ a deep, message-passing Graph Neural Network (GNN). The final representation $\mathbf{z}_i = \mathbf{h}_i^L$ and $\mathbf{z}_{ij} = \mathbf{h}_{ij}^L$ are generated by the GNN for each node and edge respectively, capturing relevant information. These representations are subsequently passed through another `MLP` to predict the force of each node. We next detail these individual steps.

**Pre-processing.** In the pre-processing layer, a dense vector representation is constructed for each node $v_i$ and edge $e_{ij}$ using an MLP denoted as $\text{MLP}_{em}$. The construction process can be described as

$$\mathbf{h}_i^0 = \text{squareplus}(\text{MLP}_{em}(\text{one-hot}(\tau_i))) \tag{3}$$

$$\mathbf{h}_{ij}^0 = \text{squareplus}(\text{MLP}_{em}(w_{ij})) \tag{4}$$

Here, `squareplus` represents an activation function. It is important to note that the node and edge embedding functions, parameterized by $\text{MLP}_{em}$, utilize separate weights. For the sake of brevity, we refer to them simply as $\text{MLP}_{em}$. Note that the node representations do not explicitly encode their ground-truth positions. Only the distances between nodes are encoded in the edge representations. This allows us to ensure *translational invariance.*

**Force prediction.** We utilize multiple layers of message-passing to facilitate communication between nodes and edges. In the $\ell^{th}$ layer, the node embeddings are updated as follows:

$$\mathbf{h}_i^{\ell+1} = \text{squareplus}\left(\mathbf{W}_{\mathcal{V}}^{\ell} \cdot \left(\mathbf{h}_i^{\ell} \,\big|\big|\, \sum_{j \in \mathcal{N}_i^{in}} \mathbf{h}_{ji}^{\ell} \,\big|\big|\, \sum_{j \in \mathcal{N}_i^{out}} \mathbf{h}_{ij}^{\ell}\right)\right) \tag{5}$$

Here, $\mathcal{N}_i^{in} = \{v_j \in \mathcal{V} \mid e_{ji} \in \mathcal{E}\}$ and $\mathcal{N}_i^{out} = \{v_j \in \mathcal{V} \mid e_{ij} \in \mathcal{E}\}$ represents the incoming and outgoing neighbors of node $v_i$ respectively. $||$ denotes the vector concatenation operation. The term $\mathbf{W}_{\mathcal{V}}^{\ell}$ signifies a layer-specific learnable weight matrix. More simply, we sum-pool the edge embeddings over both incoming and outgoing neighbors, concatenate them along with the target node's own embedding and then pass it through one linear transformation followed by non-linearity. The embedding of an edge $e_{ij}$ on node $v_i$ in the $\ell^{th}$ layer, denoted as $\mathbf{h}_{ij}^{\ell}$, is computed as:

$$\mathbf{h}_{ij}^{\ell+1} = \text{squareplus}\left(\mathbf{W}_{\mathcal{E}}^{\ell} \cdot \left(\mathbf{h}_{ij}^{\ell} \,\big|\big|\, \mathbf{h}_i^{\ell} \,\big|\big|\, \mathbf{h}_j^{\ell}\right)\right) \tag{6}$$

Similar to $\mathbf{W}_{\mathcal{V}}^{\ell}$, $\mathbf{W}_{\mathcal{E}}^{\ell}$ is a layer-specific learnable weight matrix that applies to the edge set. The message-passing process spans $L$ layers, with $L$ being a hyper-parameter. In the $L^{th}$ layer, the final node and edge representations are denoted as $\mathbf{z}_i = \mathbf{h}_i^L$ and $\mathbf{z}_{ij} = \mathbf{h}_{ij}^L$, respectively.

Finally, the pair-wise interaction force $\hat{F}_{ij}$ from particle $i$ to $j$ is predicted as:

$$\hat{F}_{ij} = \text{squareplus}(\text{MLP}_{\mathcal{V}}(\mathbf{z}_{ij})) \tag{7}$$

where, $\text{MLP}_{\mathcal{V}}$ denotes a Multilayer Perceptron with a squareplus activation function Barron (2021). One may also use ReLU as activation function, see empirical comparison at Fig. L in App. H

The force on a node is defined as the summation of all forces from incoming edges and the *reactive force* from its outgoing edges. Mathematically,

$$\hat{F}_i = \sum_{j \in \mathcal{N}_i^{in}} \hat{F}_{ji} + \sum_{j \in \mathcal{N}_i^{out}} -\hat{F}_{ij} \tag{8}$$

Here, the second term accounts for the reactive force from each outgoing edge, which is essentially the negated version of the force particle $i$ imparts on particle $j$. This inductive bias ensures Newton's third law that every action has an equal and opposite reaction. More importantly, this inductive bias implicitly ensures that the net force on the system is zero, thereby strictly ensuring momentum conservation. Formally:

**Theorem 1 (Momentum Conservation)** *In the absence of an external field,* BROGNET *exactly conserves the linear momentum of the system.*

Proof of the theorem is provided in Appendix A. Further, as demonstrated later the empirical results comparing the performance of BROGNET with a version that does not employ the momentum conservation, namely, Brownian dynamics GNN (BDGNN), suggests that the momentum conservation indeed results in better performance of the model.

**Learning the Brownian term.** While we can learn the distribution associated with the diffusion term (see Eq.( 2)), it is well-known that Brownian motion follows multivariate Gaussian distribution. The diffusion term is independent of the system topology and depends only the particle type and its attributes such as radius or friction coefficient. To learn this term, we represent the node type as a one-hot encoding, which is passed through an MLP to obtain the $\hat{\gamma}_i$ associated with each particle (see Fig. 1). This $\hat{\gamma}_i$, along with force obtained from the GNN, is substituted in the governing equation (Eq. 2) and learned in an end-to-end fashion. It is worth noting that the standard deviation of the stochastic process is given by $\sigma_i = \sqrt{2\gamma_i k_B T}$. Thus, all other terms other than $\gamma_i$ in the Eq. 2 are known apriori or are constants.

**Trajectory prediction and training.** Based on the predicted forces and $\gamma_i$s, the positions are derived using the *Euler Maruyama* integrator Braǹka & Heyes (1998). The loss on the prediction is computed using *Gaussian Negative Log-likelihood* loss, which is then back-propagated to train the model. Specifically, let $X_{i,t}$ and $\hat{X}_{i,t}$ denote the ground truth and predicted positions of particle $i$ at time $t$, respectively. The loss function is defined as follows.

$$\mathcal{L} = \frac{1}{N} \left( \sum_{i=1}^{N} \sum_{t=2}^{T} \log \max\{\hat{\sigma}_i^2, \epsilon\} + \lambda \frac{(X_{i,t} - \hat{X}_{i,t})^2}{\max\{\hat{\sigma}_i^2, \epsilon\}} \right) \tag{9}$$

Here, $\epsilon$ is a small constant, and $\lambda$ controls the relative weightage of the two terms. $\hat{\sigma}_i$ is the standard deviation associated with the position of each of the particles. The overall objective of the loss is to minimize the variance-normalized deviation from the ground truth (second term), while also ensuring a regularizing term to keep the variance low. We use Gaussian NLL loss since the stochasticity in Brownian dynamics is a Gaussian as well (recall Eq. 2).

Note that the positions, both predicted and ground-truth, are computed directly from the predicted and ground-truth forces, respectively, using *Euler Maruyama* integrator as follows.

$$X_i(t + \Delta t) = X_i(t) + F_i/\gamma_i \Delta t + \sqrt{\frac{2k_B T}{\gamma_i}} \Delta \Omega_t \tag{10}$$

where $\Delta \Omega_t$ is a random number sampled from a standard Normal distribution. It should be noted that the training approach presented here may lead to learning the dynamics as dictated by the *Euler Maruyama* integrator. Thus, the learned dynamics may not represent the "true" dynamics of the system, but one that is optimal for the *Euler Maruyama* integrator[1].

## 4 EXPERIMENTS

In this section, we benchmark the ability of BROGNET to learn Brownian dynamics directly from the trajectory and establish:

- **Accuracy:** BROGNET accurately models Brownian dynamics and outperforms baseline models.

---

[1]See App. B for details on *Euler Maruyama* integrator and other integrators for SDE.

- **Zero-shot generalizability:** The inductive architecture of BROGNET allows it to accurately generalize to systems of unseen sizes and temperatures.

The codebase and all datasets used in this work can be accessed from the repository at https://github.com/M3RG-IITD/BroGNet.

## 4.1 EXPERIMENTAL SETUP

We provide the details of the hardware and software environment in App. C.1.

• **Baselines:** To the best of the authors' knowledge, there are no prior works on learning Brownian dynamics directly from the trajectory. Thus, in order to compare the performance of BROGNET, we propose five different baselines.

(1) **Neural Network (NN):** Here, we use a feedforward MLP that takes $X_i$ vector at time $t$ as the input and predicts the positions at $t + \Delta t$ as the output.

(2) **Brownian NN (BNN):** NN in (1) does not have any physics-based inductive biases. To address this, we propose BNN, a Neural SDE that parameterizes the Brownian dynamics using a feed-forward MLP. Specifically, BNN takes $X_t$ of all the particles concatenated into a vector as input and predicts the $\hat{F}$ and $\hat{\gamma}$ as output employing a fully connected MLP. These outputs are then used in Eq. 10 to predict the updated positions. Thus, BNN employs the inductive-bias in terms of the Brownian equation of motion.

(3) **Brownian full graph network (BFGN):** The models in (1) and (2) do not exploit the topology of the system and are not permutation invariant. To this extent, we employ BFGN a graph-based baseline that takes the position as input and outputs the forces and $\gamma$ for each node. This output is then substituted in Eq. 10 to obtain the updated position. Thus, BFGN is a graph-based version of BNN. Note that BFGN employs the full graph network architecture Cranmer et al. (2020); Sanchez-Gonzalez et al. (2020), a versatile architecture that has been demonstrated for a variety of particle-based systems. It is worth noting that BFGN is not translational invariant due to the explicit use of positions as node inputs.

(4) **Brownian Dynamics GNN (BDGNN):** We also present a physics-informed baseline, BDGNN. BDGNN has all the inductive biases of BROGNET except Newton's third law. Specifically, the force on each particle $i$ is computed as:

$$F_i = \texttt{squareplus}(\texttt{MLP}_{\mathcal{V}}(\mathbf{z}_i)) \tag{11}$$

Note that BDGNN has the same architecture as BROGNET and hence allows us to evaluate the role of momentum conservation as an inductive bias in the BROGNET.

(5) **Brownian Neural Equivariant Interatomic Potentials (BNequIP):** BNequIP employs an equivariant GNNs proposed by (Batzner et al., 2022) and directly predicts the force vector as the output.

(6) **Momentum-informed BDGNN (MIBDGNN):** In order to study the effect of adding momentum conservation as a "soft-constraint", we add an additional term to the loss function of BDGNN for momentum conservation. Specifically, we add a third term to Eq. 9, namely, $\|\sum F_i\| = 0$.

• **Datasets and systems:** To compare the performance BROGNET, we selected three different systems, namely, linear spring, linear spring with binary particle types, and non-linear spring systems, all following Brownian dynamics at finite temperatures. The first is a model system with particles interacting with each other based on a linear spring and subjected to Gaussian noise. The second system comprises a non-linear spring where the force on a spring is proportional to the fourth power of displacement. The third system corresponds to a linear spring system with two different types of particles having different $\gamma$. For linear and non-linear spring systems, models are trained on $5-$spring system only, which are then evaluated on other system sizes. In case of linear spring with binary particle types, models are trained on $10-$spring system and evaluated on other system sizes. The details of all the systems to generate the ground truth data are provided in App. C. Further, the detailed data-generation procedure is given in App. D.

• **Evaluation Metric:** To quantify performance, we use the following metrics.

**Position error:** The position error is the normalized Euclidean distance between the ground truth and predicted positions. Mathematically, let $(x, y, z)$ and $\hat{x}, \hat{y}, \hat{z}$ be the ground truth and predicted positions, respectively, of a given particle. Furthermore, let $\sigma_{x/y/z}$ be the ground-truth standard deviation of the positions of the particle. The error is, therefore:

$$PE = \left( \sum_{c \in \{x,y,z\}} \left( \frac{c - \hat{c}}{\sigma_c} \right)^2 \right)^{1/2} \tag{12}$$

**Brownian error:** The Brownian error quantifies the difference between the standard deviation

estimated by the neural network and that of the ground truth using the RMSE metric.

**Trajectory roll-out error:** The above two metrics individually look at the errors in positions and the standard deviation. In this metric, we holistically compare the distance between the distribution of trajectories in the ground truth against the ones predicted by BROGNET. Specifically, given a Brownian system, we simulate 100 ground-truth trajectories across 100 time steps over each particle in the system. Similarly, we predict the trajectories using BROGNET. We next compute the average distance between the ground-truth distribution and the predicted one using KL-divergence across all particles. KL-divergence is defined as follows:

$$D_{\text{KL}}(\hat{\mathcal{X}} \| \mathcal{X}) = \sum_{x \in \hat{\mathcal{X}}} \hat{\mathcal{X}}(x) \log \left( \frac{\hat{\mathcal{X}}(x)}{\mathcal{X}(x)} \right) \tag{13}$$

Here, $\mathcal{X}$ and $\hat{\mathcal{X}}$ represent the distributions over the ground truth and predicted trajectories, respectively. Both these distributions are univariate normal distributions since the stochastic component in Brownian dynamics is normally distributed (Eq. 2). Hence, Eq. 13 reduces to the following:

$$D_{\text{KL}}(\hat{\mathcal{X}} \| \mathcal{X}) = \log \frac{\sigma_1}{\sigma_0} + \frac{\sigma_0^2 + (\mu_o - \mu_1)^2}{2\sigma_1^2} - \frac{1}{2} \tag{14}$$

Here, $\sigma_0$ and $\mu_0$ are the standard-deviation and mean of distribution $\hat{\mathcal{X}}$ and similarly $\sigma_1$ and $\mu_1$ denote the standard deviation and mean of $\mathcal{X}$.

• **Model architecture and training setup:** We use 10000 data points generated from 100 trajectories with random initial conditions to train all the models. Detailed training procedures and hyper-parameters are provided in App. E. For all versions of the graph architectures, MLPs are two layers deep. All models were trained till the decrease in loss saturates to less than 0.001 over 100 epochs. The model performance is evaluated on a forward trajectory, a task it was not explicitly trained for, of 0.1s. To compute the evaluation metrics, all the results are obtained over trajectories generated from 100 different initial conditions. Further, to evaluate the zero-shot generalizability, the models trained on 5 spring systems are evaluated on 50, and 500 spring systems. Further, the models trained at 1 unit temperature are evaluated at 10 and 100 unit temperatures.

## 4.2 COMPARISON WITH BASELINES

Fig. 2 shows the trajectory error, Brownian error, and the position error and the geometric mean of these errors over the trajectory for all the baselines, namely, NN, BNN, BFGN, BDGNN, MIBDGNN, and BNequIP in comparison to BROGNET for linear 5−spring system. We observe that BROGNET and MIBDGNN significantly outperform all other baselines in terms of overall evaluation metrics. Specifically, we observe that the BROGNET outperforms BDGNN, which differs from the BROGNET only in terms of momentum conservation. This suggests that the momentum conservation bias added BROGNET significantly enhances the performance of the model. Interestingly, we observe that the MIBDGNN significantly outperforms BROGNET. This shows that the additional regularizer in the loss function enables improved learning of the dynamics.

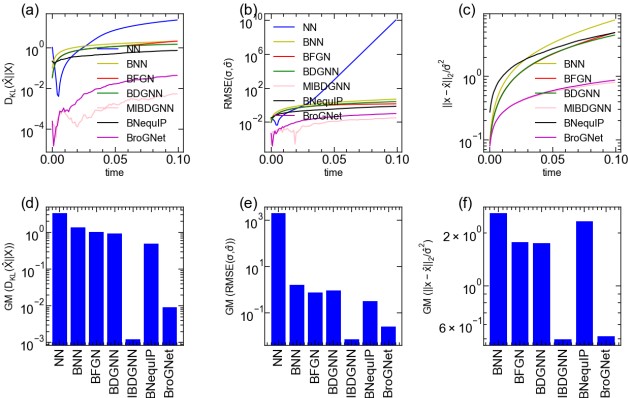

Figure 2: Comparison of BROGNET on a linear 5−spring system against baselines, NN, BNN, BFGN, BDGNN, MIBDGNN and BNequIP using (a) trajectory roll-out error, (b) Brownian error, and (c) position error. All models are trained using 10000 data points. The plots are obtained based on 1000 forward simulations by the trained model from 100 initial conditions, each evaluated with 10 random seeds random. Additionally, the geometric mean of (d) trajectory roll-out error, (e) Brownian error, and (f) position error is shown in the bottom subplot.

Figure 3 shows the performance of BROGNET on non-linear 5−spring system, and Figure 4 shows the performance on binary linear 10−spring systems, respectively. As in the case of the linear spring system, we observe that BROGNET outperforms all other baselines in learning the Brownian dynamics for non-linear spring systems and binary linear spring systems. However, we note that

the difference between the performance of BROGNET with other baselines is lower in binary spring systems. Nevertheless, these results suggest the ability of BROGNET to learn the dynamics of complex multi-particle systems interacting with each other based on non-linear interaction forces.

## 4.3 ZERO-SHOT GENERALIZABILITY

Now, we evaluate the zero-shot generalizability of the learned models to unseen system sizes and temperatures. To this extent, we use the learned models, BFGN, BDGNN, MIBDGNN, BNequIP and BROGNET trained on N = 5 and T = 1 units. First, these models are evaluated on system sizes of 50 and 500 springs, up to two orders of magnitude larger than those used in the training data. Figure 5(a-c) shows the trajectory rollout error of BFGN, BDGNN, MIBDGNN, BNequIP and BROGNET on 5−, 50−, and 500− spring systems, respectively. Interestingly, we observe all four models exhibit inductivity to larger system sizes, thanks to the GNN. Further, we observe that the trajectory rollout error for all the models is comparable with the increasing system size, even when evaluated on a system size of 500. All the error met-

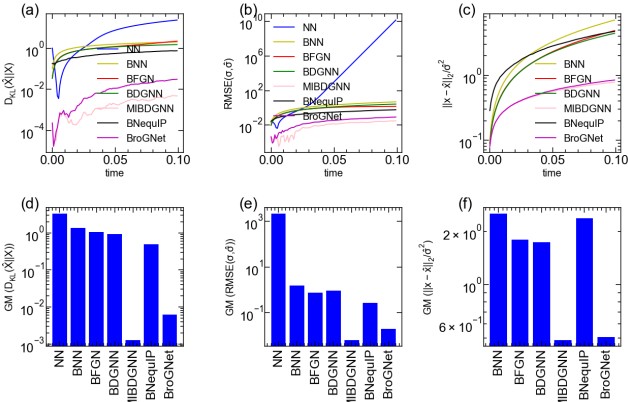

Figure 3: Comparison of BROGNET on a non-linear 5−spring system against baselines NN, BNN, BFGN, BDGNN, MIBDGNN and BNequIP using (a) trajectory roll-out error, (b) Brownian error, and (c) position error. All models are trained using 10000 data points. The plots are obtained based on 1000 forward simulations by the trained model from 100 initial conditions, each evaluated with 10 random seeds. Additionally, the geometric mean of (d) trajectory roll-out error, (e) Brownian error, and (f) position error are shown in the bottom subplot.

rics on all three types of systems linear spring, binary linear spring, and non-linear spring are included in the App. F. These results suggest that the BROGNET can be trained on small datasets and generalized to arbitrarily large system sizes for performing inference on the Brownian dynamics of the system. It should be noted that both NN and BNN, due to their architecture (feed-forward MLP), are not inductive and do not allow inference on larger system sizes.

Another important aspect of Brownian dynamics is the ability to simulate the learned systems corresponding to different thermodynamic conditions, namely, different temperatures. Figure 5(e,f) shows the trajectory roll-out error of linear spring systems with five particles at temperatures 10 and 100 units, respectively. We again observe that BROGNET and MIB-DGNN outperforms BFGN, BDGNN, and BNequIP with very low errors on the roll-out. This suggests that both the stochastic Brownian terms and the deterministic force term learned by the BROGNET are accurate and scalable to large system sizes and different thermodynamic conditions.

## 4.4 MOMENTUM CONSERVATION

Note that the Brownian motion has a net-zero acceleration, so the momentum remains conserved. To evaluate this, we compute the total force experienced by each particle during the

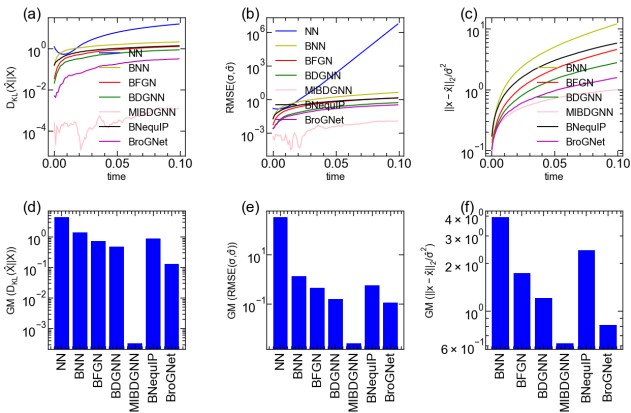

Figure 4: Comparison of BROGNET on a binary linear 10−spring system against baselines using (a) trajectory roll-out error, (b) Brownian error, and (c) position error. All models are trained using 10000 data points. The plots are obtained based on 1000 forward simulations by the trained model from 100 initial conditions, each evaluated with 10 random seeds random. Additionally, the geometric mean of (d) trajectory roll-out error, (e) Brownian error, and (f) position error are shown in the bottom subplot.

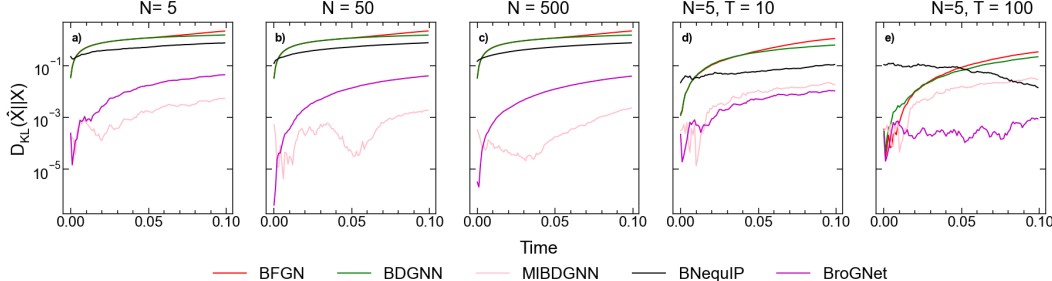

Figure 5: Trajectory roll-out error of BFGN, BDGNN, MIBDGNN, BNequIP and BROGNET trained on N = 5 and T =1 to unseen system sizes (N = 50, and 500) and unseen temperatures (T = 10 and 100) on the linear spring system. The plots are obtained based on 100 forward simulations by the trained model with random initial conditions.

dynamics predicted by the models. Figure 6 shows the geometric mean of the total force experienced by the system. Interestingly, we see that all the models except BROGNET predict finite force, while BROGNET predicts force close to zero. Although MIBDGNN exhibits lower error than other models, BROGNET outperforms MIBDGNN in this metric; as expected from a hard constraint vs. soft constraint. This shows that the additional inductive bias of momentum conservation enforced in the architecture of BROGNET contributes to its improved performance.

## 4.5 DATA EFFICIENCY

We also evaluated the data efficiency of BROGNET compared to the baselines. Specifically, we evaluated the performance of BROGNET trained with dataset size varying as 100, 500, 1000, 5000, and 10000 and compared it with the baselines BFGN, BDGNN, MIBDGNN and BNequIP, trained with the respective dataset. The results shown in Fig. 7 suggest that BROGNET can learn efficiently from a small dataset size compared to the baselines, which could be attributed

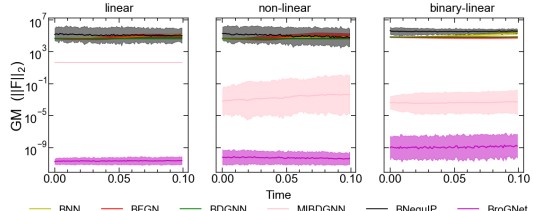

Figure 6: Momentum Error of linear, non-linear, and binary-linear spring based on 100 forward simulations by the trained model with 10 random initial conditions.

to the momentum conservation bias in the system.

## 5 CONCLUSION

We demonstrate a graph-based neural SDE framework that can learn Brownian dynamics directly from the data. We demonstrate the approach to linear and non-linear springs and binary linear spring systems subjected to a stochastic dynamic term and following Brownian dynamics. In order to compare the performance of BROGNET, we proposed several baselines, namely, NN, BNN, BFGN, BDGNN, MIBDGNN, and BNequIP. Note that these baselines were proposed due to the paucity

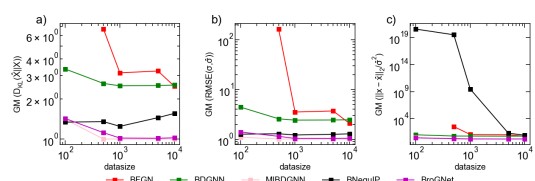

Figure 7: Geometric mean of (a) trajectory error, (b) Brownian error, and (c) position error of BFGN, BDGNN, MIBDGNN, BNEQUIP, and BROGNET trained on varying dataset size. All the results are evaluated on 1000 forward simulations; 100 initial conditions, each evaluated with 10 random seeds.

of baselines in the literature. We show that the BROGNET and MIBDGNN architectures, with their unique momentum conservation, significantly outperforms all other baselines. Specifically, we show that MIBDGNN exhibits superior performance in learning the dynamics due to the additional term in the loss function, while BROGNET exhibits superior momentum conservation due to the hard architectural constraint. We also demonstrate the zero-shot generalizability of the learned model to unseen system sizes and unseen temperatures. These results suggest that BROGNET presents a robust framework to learn Brownian dynamics from small amounts of data.

**Limitations and future works:** This work focuses on Brownian dynamics, which is an over-damped limit of the Langevin equation. How does the model perform in the case of a Langevin equation where the acceleration is non-zero? How can the model be extended to other SDEs? We aim to explore these questions in our future work. Moreover, the present architecture cannot learn the dynamics in the presence of external fields. Extending the model to the dynamics of real-world systems, including experimental datasets, is also an outstanding problem.

ACKNOWLEDGMENTS

NMAK and SR acknowledge the financial support for this research provided by Intel and Google Research Scholar Award. The authors thank the IIT Delhi HPC facility for providing computational and storage resources. SB acknowledges the financial support from the Prime Minister's Research Fellowship (PMRF), Ministry of Education, Government of India.

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

## A APPENDIX

### A PROOF OF THM. 1

PROOF. Consider a system without any external field where the dynamics is governed only by the internal deterministic forces, for instance, a $n$-spring system and stochastic force terms representing the interaction with the environment, for example, spring-particle systems in fluid system with the stochastic term representing the collision of the particles with the atoms of the fluid environment. Note that the deterministic term consisting of the forces is also termed as the drift term as it contributed to the overall drift of the system, if any. Accordingly, the stochastic term is referred to as the diffusion term. Since, the acceleration of each of the particles is zero, the net force on the system should be zero, that is, the total linear momentum of the system due to the internal deterministic forces remain conserved. The conservation of linear momentum for this system implies that $\sum_{i=1}^{n} \hat{F}_i = 0$. Assume that the total error between the predicted and actual forces of all the particles at time $t$ to be $\hat{\varepsilon}_t$. To prove that BROGNET conserve the linear momentum exactly, it is sufficient to prove that $\hat{\varepsilon}_t = 0$, since the stochastic term has zero mean. Consider,

$$\hat{\varepsilon}_t = \sum_{i=1}^{n} \hat{\varepsilon}_{i,t} = \sum_{i=1}^{n} (\hat{F}_i - F_i) = \sum_{i=1}^{n} \hat{F}_i - \sum_{i=1}^{n} F_i \tag{15}$$

Note that in Brownian dynamics, which is the over damped limit of Langevin equation (Eq. 1), the acceleration of each particle is assumed to be zero. Thus, without loss of generality, $\sum_{i=1}^{n} F_i = F_i = 0$, in the absence of an external field. Hence,

$$\hat{\varepsilon}_t = \sum_{i=1}^{n} \hat{\varepsilon}_{i,t} = \sum_{i=1}^{n} \hat{F}_i = \sum_{i=1}^{n} \left( \sum_{j \in \mathcal{N}_i^{in}} \hat{F}_{ji} + \sum_{j \in \mathcal{N}_i^{out}} -\hat{F}_{ij} \right) = 0 \qquad \square$$

Thus, the sum of the predicted force of the system remains zero leading to the conservation of linear momentum of the system.

### B EULER MARUYAMA INTEGRATOR

Consider a system of $N$ particles having masses $M$ and interacting with each other through an interaction potential $U(X(t))$, where $X(t)$ represents the position vectors of all the particles at time $t$, that constitute a time-dependent random variable. The Taylor series expansion of the position of a particle $i$ at time $t + \Delta t$ is given by

$$X_i(t + \Delta t) = X_i(t) + \Delta t \dot{X}_i(t) + \Delta t^2 \ddot{X}_i(t) + \ldots \tag{16}$$

In the over-damped limit of Langevin dynamics, namely, Brownian dynamics, acceleration and higher order terms are assumed to be zero. Thus, substituting for velocity given by Eq. 2 in Eq. 16, we get

$$X_i(t + \Delta t) = X_i(t) + F_i/\gamma_i \Delta t + \sqrt{\frac{2k_B T}{\gamma_i}} \Delta \Omega_t \tag{17}$$

where $\Delta \Omega_t$ is a random number sampled from a standard Normal distribution. Note that the Euler-Maruyama is the simplest form of a stochastic numerical integrator. Further, higher order integrals can also be employed to integrate the SDEs.

### C EXPERIMENTAL SYSTEMS

#### C.1 SIMULATION ENVIRONMENT

• **Simulation environment.** All the training and forward simulations are carried out in the JAX environment Schoenholz & Cubuk (2020). The graph architecture is implemented using the jraph package Godwin* et al. (2020). All the codes related to dataset generation and training are available in https://github.com/M3RG-IITD/BroGNet.
**Software packages:** numpy-1.20.3, jax-0.2.24, jax-md-0.1.20, jaxlib-0.1.73, jraph-0.0.1.dev0
Hardware: Memory: 16GiB System memory, Processor: Intel(R) Core(TM) i7-10750H CPU @ 2.60GHz.

## C.2 LINEAR $n$-SPRING SYSTEM

In this system, $n$-point masses are connected by elastic springs that deform linearly with extension or compression. Note that similar to a pendulum setup, each mass $m_i$ is connected only to two masses $m_{i-1}$ and $m_{i+1}$ through springs so that all the masses form a closed connection. Thus, the deterministic force experienced by each mass $i$ due to a spring connecting $i$ to its neighbor $j$ is $F_{ij} = -k(||X_i - X_j|| - R_{ij})$, where $R_{ij}$ is the equilibrium length of the spring and $k$ represent the un-deformed length and the stiffness, respectively, of the spring.

## C.3 NON-LINEAR $n$-SPRING SYSTEM

This system is similar to a linear $n$-spring system with the difference that the spring force is non-linear. Specifically, the deterministic force experienced by each mass $i$ due to a spring connecting $i$ to its neighbor $j$ is $F_{ij} = -k(||X_i - X_j|| - R_{ij})^3$, where $R_{ij}$ is the equilibrium length of the spring and $k$ represent the un-deformed length and the stiffness, respectively, of the spring.

## C.4 BINARY LINEAR $n$-SPRING SYSTEM

This system is similar to a linear $n$-spring system with the difference that the system consists of two different types of particles with different masses, friction coefficient, and $\gamma$. Thus, the deterministic force experienced by each mass $i$ due to a spring connecting $i$ to its neighbor $j$ remains the same as $F_{ij} = -k(||X_i - X_j|| - R_{ij})$, where $R_{ij}$ is the equilibrium length of the spring and $k$ represent the un-deformed length and the stiffness, respectively, of the spring. However, the stochastic part varies depending on the type of the particle.

## D DATASET GENERATION

All the datasets are generated using the known deterministic forces of the systems, along with the stochastic, as described in Section C and Eq.2. For each system, we create the training data by performing forward simulations with 100 random initial conditions. For both linear and nonlinear $n-$spring system, $n = 5$ is used for generating the training data. For the binary system, a 10 particle system is used for generating the data with the ratio of particle type 1 to type 2 as 3:7. A timestep of $10^{-3}s$ is used to integrate the equations of motion for all the systems. The Euler-Maruyama algorithm is used to integrate equations of motion due to its ability to handle stochastic differential equations. The details of the parameters used for each of the systems are given below.

● **Linear $n$-spring system**

| Parameter | Value |
|---|---|
| Mass ($M$) | 1 unit |
| Stiffness ($k$) | 1 unit |
| Damping constant ($\zeta$) | 1 unit |
| $k_B T$ | 1 unit |
| Equilibrium length of the spring ($R_{ij}$) | 1 unit |

● **Nonlinear $n$-spring system**

| Parameter | Value |
|---|---|
| Mass ($M$) | 1 unit |
| Stiffness ($k$) | 1 unit |
| Damping constant ($\zeta$) | 1 unit |
| $k_B T$ | 1 unit |
| Equilibrium length of the spring ($R_{ij}$) | 1 unit |

● **Binary linear $n$-spring system**

| Parameter | Value |
|---|---|
| Mass ($M$) | 1 unit |
| Stiffness ($k$) | 1 unit |
| ($\gamma_1$) | 1 unit |
| ($\gamma_2$) | 2 unit |
| $k_B T$ | 1 unit |
| Equilibrium length of the spring ($R_{ij}$) | 1 unit |
| Ratio of particle type1 to type2 | 30:70 |

From the 100 simulations for the systems obtained from the rollout starting from 100 random initial conditions, 100 data points are extracted per simulation, resulting in a total of 10000 data points. For training, we create several sets of trajectories with positions from two consecutive timesteps $t$ and $t + \Delta t$, where the input for the model is the positions of all the particles at $t$ and the output against which the model is trained is the position of all the particles at $t + \Delta t$ employing the loss function as defined in Eq. 9. Note that in the present work, we use only trajectories of length one timestep only. The trained models are evaluated on 1000 trajectories generated from 100 different initial conditions with each initial condition simulated for 10 random seeds, all of which are unseen during the training. The trajectories considered are of 100 timesteps, that is, 0.1 s. Note that this is not a task the models are trained for; the training was done only on one timestep trajectories. All the error metrics presented in the work are averaged over the 1000 trajectories for each of the models and casees unless specified otherwise.

For zero-shot generalizability, for each case of different system sizes and temperatures, we directly evaluate the trained models on 1000 trajectories. No fine tuning or additional training is performed for any of the different system sizes or temperatures.

## E  TRAINING DETAILS AND HYPER-PARAMETERS

The detailed procedures followed for the training of the models and the hyperparameters employed for each of the models, identified based on good practices, are provided in this section.

### E.1  TRAINING DETAILS

The training dataset is divided in 80:20 ratio randomly, where the 80% is used for training and 20% is used as the validation set for hyperparametric optimization. Further, the trained models are tested on its ability to predict statistically equivalent trajectory, a task it was not trained on. Specifically, the systems are evaluated on a $0.1s$ long trajectory, that is $100\times$ larger than the training trajectory. The error metrics are computed based on 1000 different trajectories from 100 initial conditions, each run with 10 random seeds. All models are trained till the losses saturated. A learning rate of $10^{-3}$ was used with the Adam optimizer for the training. Detailed hyperparameters employed for all the models are provided next.

### E.2  HYPER-PARAMETERS

The hyper-parameters used for different models, namely, NN, BNN, BFGN, BDGNN, and BROGNET are given in the tables below.

•**NN, BNN**

| Parameter | Value |
|---|---|
| Hidden layer neurons (MLP) | 16 |
| Number of hidden layers (MLP) | 2 |
| Activation function | squareplus |
| Optimizer | ADAM |
| Learning rate | $1.0e^{-3}$ |
| Batch size | 20 |

•**BFGN**

| Parameter | Value |
|---|---|
| Node embedding dimension | 8 |
| Edge embedding dimension | 8 |
| Hidden layer neurons (MLP) | 16 |
| Number of hidden layers (MLP) | 2 |
| Activation function | squareplus |
| Number of layers of message passing | 1 |
| Optimizer | ADAM |
| Learning rate | $1.0e^{-3}$ |
| Batch size | 20 |

• **BDGNN, BROGNET**

| Parameter | Value |
|---|---|
| Node embedding dimension | 5 |
| Edge embedding dimension | 5 |
| Hidden layer neurons (MLP) | 5 |
| Number of hidden layers (MLP) | 2 |
| Activation function | squareplus |
| Number of layers of message passing | 1 |
| Optimizer | ADAM |
| Learning rate | $1.0e^{-3}$ |
| Batch size | 20 |

• **BNequIP**

| Parameters | Value |
|---|---|
| graph_net_steps | 2 |
| use_sc | True |
| nonlinearities | {'e': 'raw_swish', 'o': 'tanh'} |
| n_element | 1 |
| hidden_irreps | '128x0e + 64x1e + 4x2e' |
| sh_irreps | '1x3e + 1x0e' |
| num_basis | 8 |
| r_max | 2.5 |
| radial_net_nonlinearity | 'raw_swish' |
| radial_net_n_hidden | 64 |
| radial_net_n_layers | 1 |
| shift | 0 |
| scale | 1 |
| n_neighbors | 10. |
| scalar_mlp_std | 5. |
| Optimizer | ADAM |
| Learning rate | $1.0e^{-2}$ |
| Batch size | 100 |

## F  ZERO-SHOT GENERALIZABILITY

Here, we evaluate the zero-shot generalizability of BROGNET in comparison to BFGN, BDGNN, MIBDGNN, and BNequIP for linear, non-linear, and binary linear $n-$ spring systems. Figures H I and J show the trajectory, Brownian, and position error, respectively, of BFGN, BDGNN, MIBDGNN, BNequIP, and BROGNET evaluated on 50, 500, and 5000 particle systems. Note for linear and non-linear springs, the models are trained on $5-$particle systems, while for binary linear spring system, the models are trained on $10-$particle systems. We observe that BROGNET significantly outperforms other models. Further, Fig. K shows the Inference time w.r.t number of particle for BFGN, BDGNN, MIBDGNN, BROGNET and BNEQUIP.

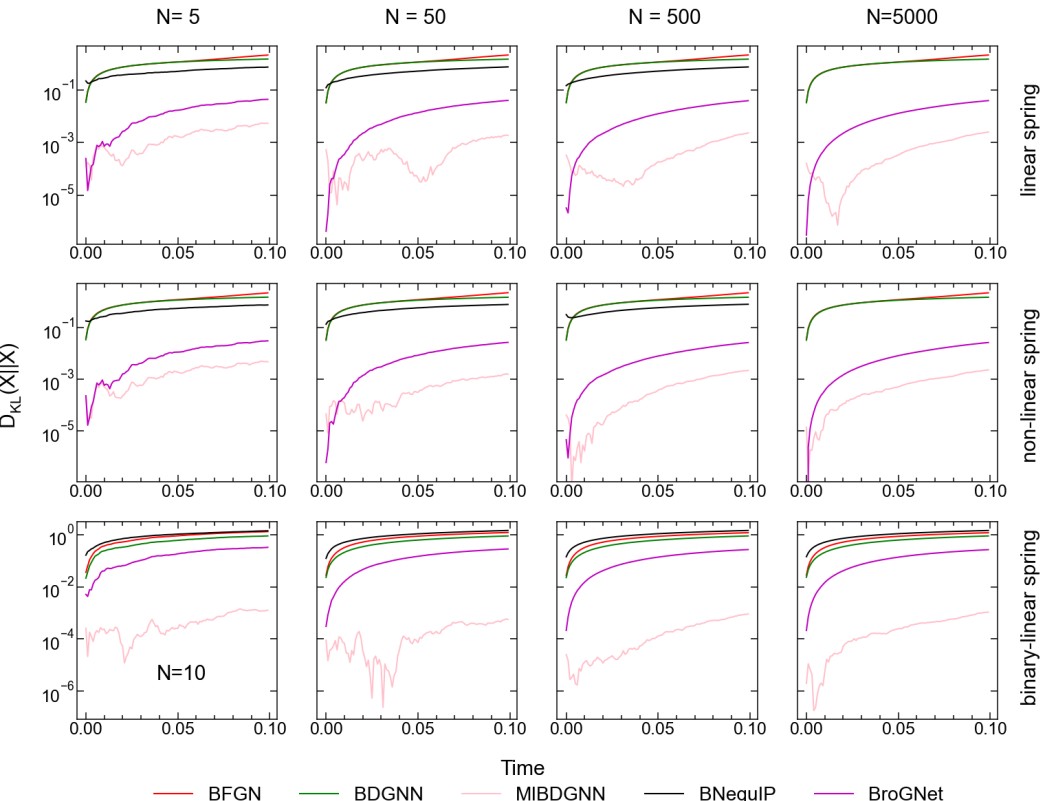

Figure H: Rollout error of BFGN, BDGNN, MIBDGNN, BNequIP, and BROGNET tested on N = 50, 500, 5000 for linear (row 1), non-linear spring (row 2), and binary linear spring (row 3) systems evaluated on 1000 forward trajectories. Note that for linear and non-linear systems, the models are trained on 5−particle systems whereas for binary system, the models are trained on 10−particle system.

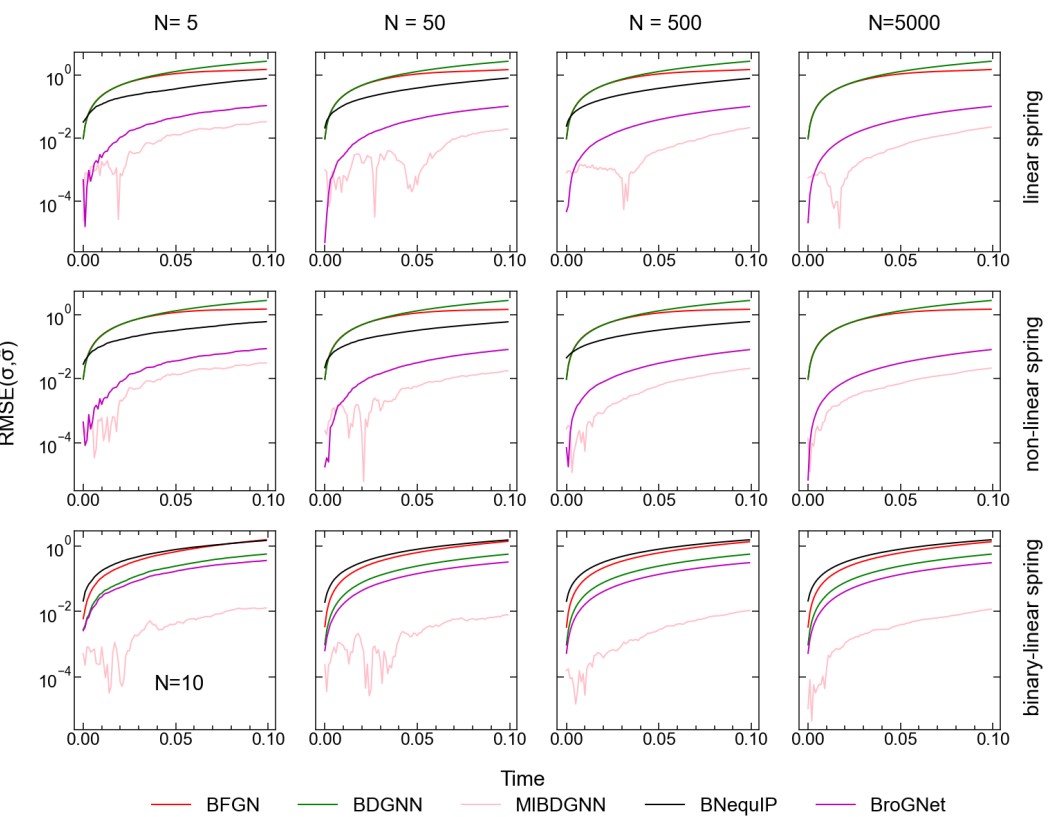

Figure I: Brownian error of BFGN, BDGNN, MIBDGNN, BNequIP, and BROGNET trained on N = 5 system and tested on N = 50, 500, 5000 for linear (row 1), non-linear spring (row 2), and binary linear spring (row 3) systems evaluated on 1000 forward trajectories.

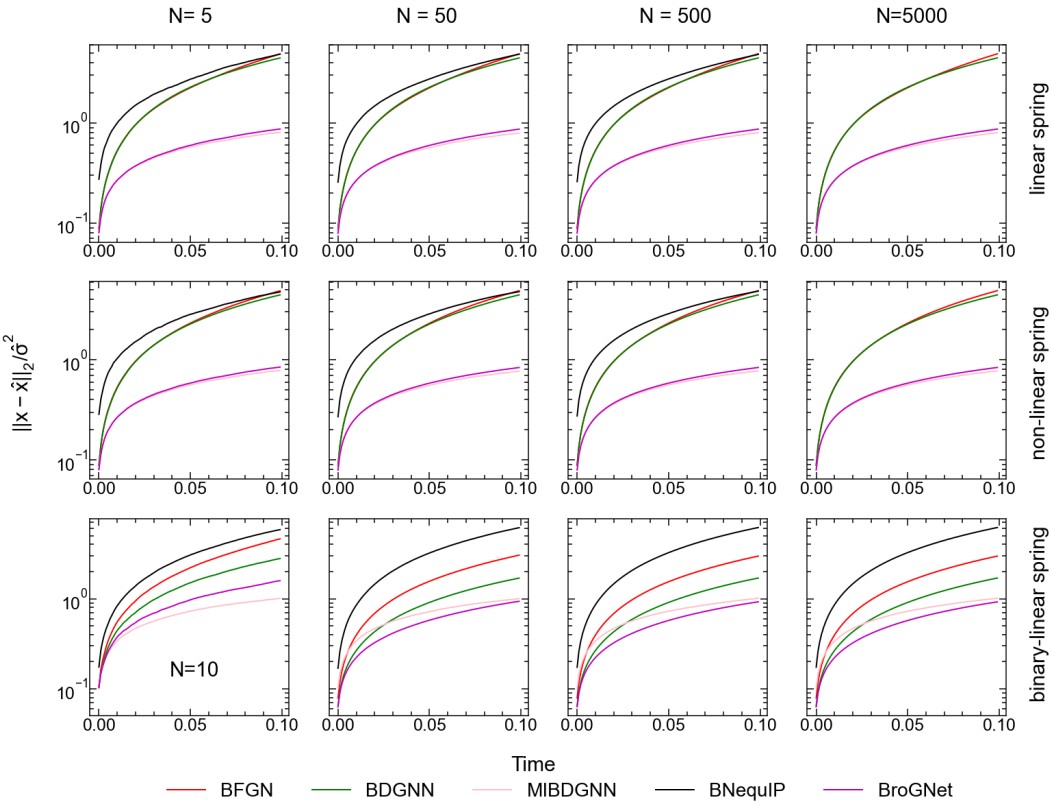

Figure J: Position error of BFGN, BDGNN, MIBDGNN, BNequIP, and BROGNET trained on N = 5 system and tested on N = 50, 500, 5000 for linear (row 1), non-linear spring (row 2), and binary linear spring (row 3) systems evaluated on 1000 forward trajectories.

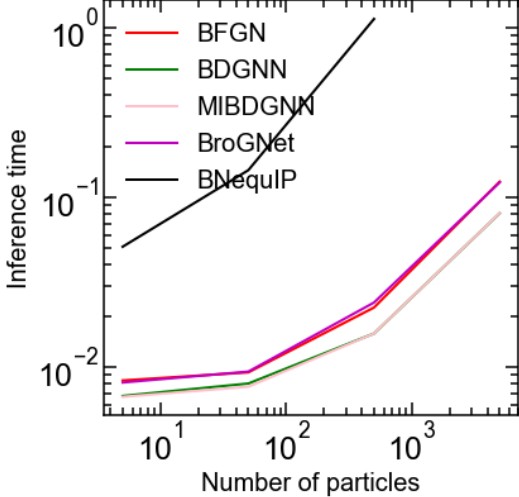

Figure K: Inference time w.r.t number of particle for BFGN, BDGNN, MIBDGNN, BROGNET and BNEQUIP. Note that in this the forward simulation time is computed as an average of 30 trajectories.

| Models | Training time (in sec) | Forward Simulation time (in sec) |
|---|---|---|
| NN | 391 | 0.012 |
| BNN | 635 | 0.222 |
| BFGN | 1755 | 0.642 |
| BDGNN | 1934 | 0.411 |
| BNequIP | 27535 | 21.319 |
| BROGNET | 2126 | 0.506 |

Table A: Training time for 10,000 epochs and forward simulation time for 100 timesteps of NN, BNN, BFGN, BDGNN, NequIP, and BROGNET on linear 5 spring system. Note that the forward simulation time is computed as the average of 100 initial condition with each initial condition simulated for 10 different random seeds (altogether, an average of 1000 trajectories).

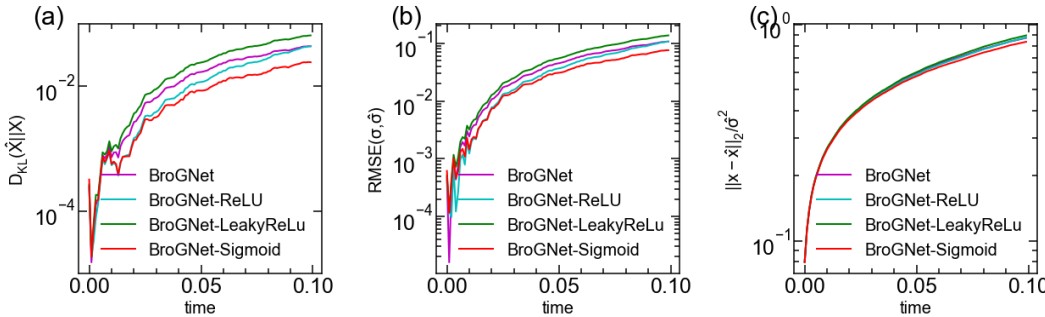

Figure L: Comparison of activation function SquarePlus, ReLU, LeakyReLU and Sigmoid on linear $5-$ spring system with BROGNET. All the results are evaluated based on 1000 forward simulations based on 100 initial conditions, each evaluated with 10 random seeds.

## G    TRAJECTORY VISUALIZATION

Visualizations of the trajectories of linear $n$-spring systems using ground truth and BROGNET is included as a video in the link: https://github.com/M3RG-IITD/BroGNet/tree/main/videos. Note that BROGNET is trained on a $5-$particle system.

## H    SQUAREPLUS, RELU, LEAKYRELU VS SIGMOID

Here, we evaluate BROGNET performance with respect to different activation functions: SquarePlus, ReLU, LeakyReLU and Sigmoid. From Fig. L, we observe that all the activation functions give comparable performance. We used SquarePlus in order to potentially extend BrogNet to learn energy instead of force in future work. In this case, the activation function needs to be double differentiation.

## I    GENERALIZATION TO UNSEEN $\gamma$

Here, we evaluate BROGNET performance with respect to different gamma values: 0.1, 0.6, 0.8, 1.0, 1.2, 1.4, and 10. As $\gamma$ increases performance improves. This is due to the fact that for a higher value of $\gamma$, there will be a smaller standard deviation for the stochastic term.

## J    IMPORTANCE OF GRAPH TOPOLOGY

Here, we evaluate BROGNET when there is correct graph topology vs BROGNETRND when there is random connection between senders and receivers. From Fig. N it is clear that performance is significantly high with correct graph topology

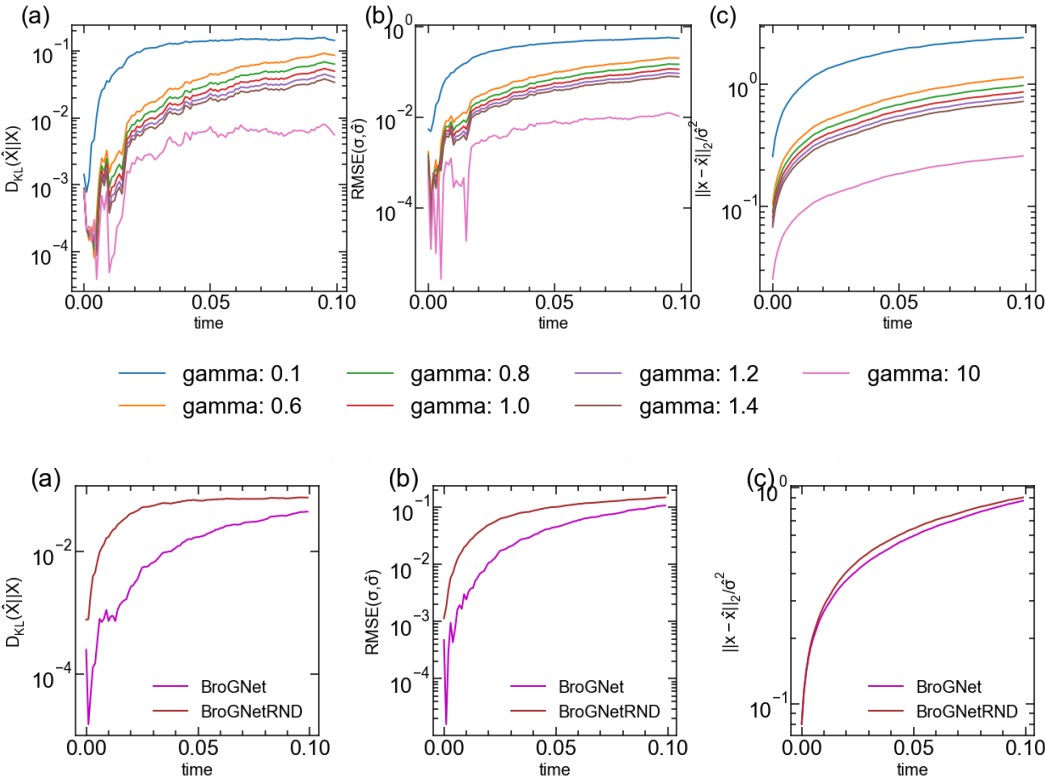

Figure N: BROGNET vs BROGNETRND architecture

## K    PERFORMANCE W.R.T. HYPERPARAMETERS

Here, we evaluate the performance of BROGNET w.r.t. number of layers of message passing [1, 2, 3] and the number of hidden layer neurons [5, 8, 16, 32]. In the paper for BROGNET we choose the number of layers of message passing = 1 and the number of hidden layer neurons = 5. For other architectural details, check Appendix E

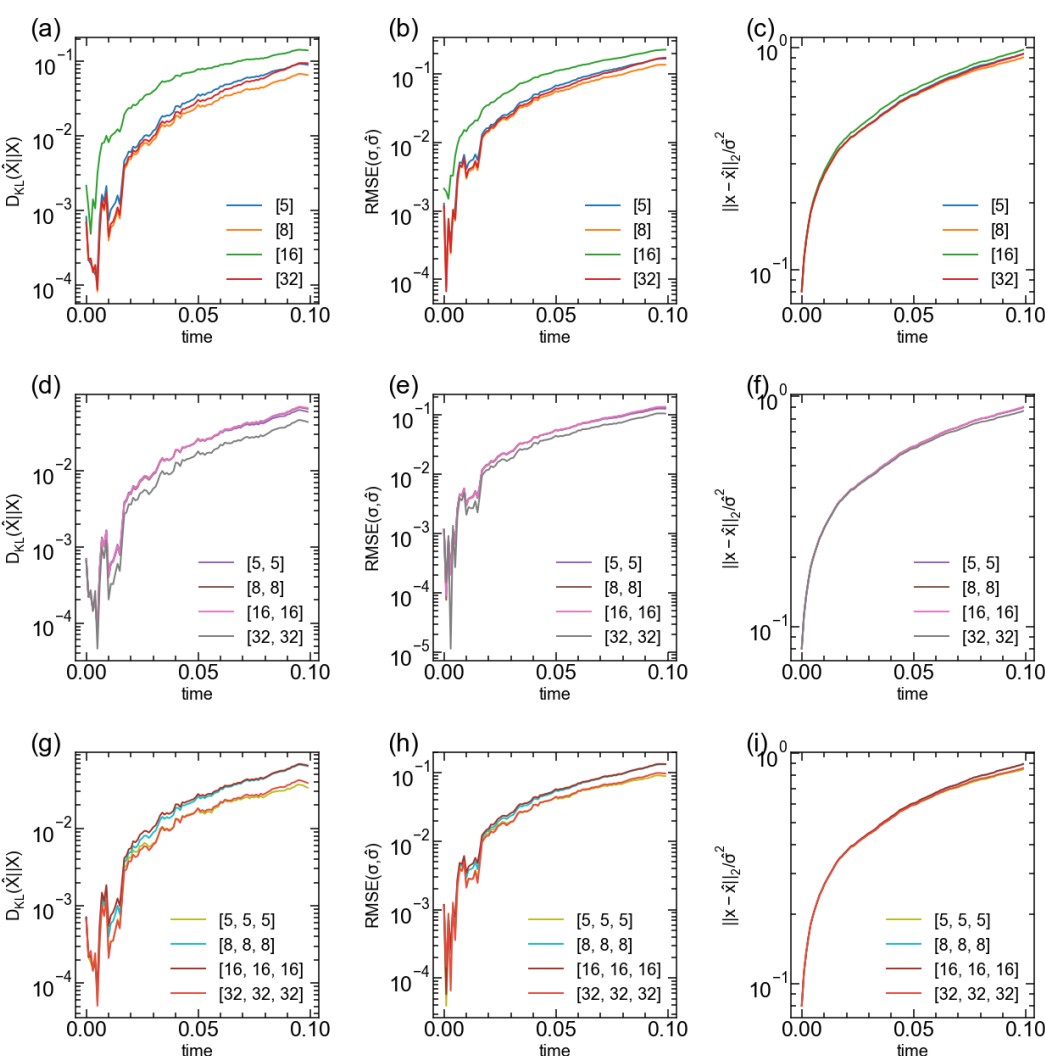

Figure O: Performance of BROGNET architecture w.r.t. hyperparameters

