demonstrate that momentum conservation is an important feature that can significantly enhance the performance of such models. We also demonstrate the zero-shot generalizability of the learned model to unseen system sizes and unseen temperatures. These results suggest that BROGNET presents a robust framework to learn Brownian dynamics from small amounts of data.

**Limitations and future works:** This work focuses on Brownian dynamics, which is an over-damped limit of the Langevin equation. How does the model perform in the case of a Langevin equation where the acceleration is non-zero? How can the model be extended to other SDEs? We aim to explore these questions in our future work. Further, the present work does not analyze the effect of different integrators. This can also be pursued as part of future work.

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

two activation functions: SquarePlus and ReLU. From Fig. K we can say one may also use ReLU for training.