# OpenReview forum: "BroGNet: Momentum-Conserving Graph Neural Stochastic Differential Equation for Learning Brownian Dynamics"
_ICLR.cc/2024/Conference — ICLR 2024 poster_

### Official Review · Reviewer_Va4f · 2023-10-26

**Soundness:** 3 good
**Presentation:** 3 good
**Contribution:** 3 good
**Rating:** 6
**Confidence:** 3

**Summary:**

## Summary

The paper proposes a Graph Neural Network (GNN)-based method for simulating the over-damped limit of Langevin dynamics, which reduces to Brownian dynamics when no acceleration is present. Contributions include the use of additional MLPs for predicting random diffusion and decoding edge latent as interacting forces to enforce linear momentum conservation. However, the paper has several significant shortcomings, including a lack of motivation, limited benchmarking, and insufficient definition and references. Due to these issues, I recommend rejecting the paper.

## Detailed Comments

### 1. Lack of Motivation and Benefits

The paper does not provide adequate motivation for replacing traditional simulation techniques for the over-damped Langevin system with GNNs. Given that the dataset is generated from simulators that can easily model such systems, the authors need to justify the utility of their approach. This could be in the form of improved simulation speed or to-reality accuracy, although the latter would require validation beyond simulated data.

### 2. Lack of Definitions and References

- In Figure 1, the term "ohe(type)" is used without definition or context, making it unclear to the reader.
- The "squareplus" activation function is mentioned but not cited.

### 3. Limited Benchmarking

The paper restricts its experiments to simple spring systems, providing only a narrow validation of its methodology. Prior work in this domain typically includes experiments on 3-4 different datasets to establish the method's applicability.

## Conclusion

While the paper introduces a GNN-based method for simulating over-damped Langevin dynamics, it suffers from multiple critical flaws, including a lack of clear motivation, insufficient references, and limited benchmarking. These drawbacks severely compromise the paper's value and applicability. Therefore, I recommend rejecting this submission.

The authors should consider submitting to a workshop on this specific topic or, including more datasets to show the method's capability.

**Strengths:**

Small contributions like: decoding edge as force so linear momentum is conserved.

**Weaknesses:**

See above.

**Questions:**

How hard is it to simulate Brownian motions? If not hard, w.r.t. simulation time or numerical or modeling challenges, there is no point in replacing it with NN?

If otherwise, please show the challenges in your draft.

---

> ### Author Response · Authors · 2023-11-19
> **Response to Reviewer Va4f**
>
> *Q1. Lack of Motivation and Benefits: The paper does not provide adequate motivation for replacing traditional simulation techniques for the over-damped Langevin system with GNNs. Given that the dataset is generated from simulators that can easily model such systems, the authors need to justify the utility of their approach. This could be in the form of improved simulation speed or to-reality accuracy, although the latter would require validation beyond simulated data.*
>
> **Response**: There is a large family of works based on machine learning approaches in general and graph neural networks in particular that aims at learning the dynamics of physical systems. Such approaches can be used for system identification, that is, to learn the quantities such as noise, interparticle forces and other features of the system directly from the trajectory. Thus, these approaches can be used for "discovering" interaction laws and dynamics and also to infer them in a realistic manner for future timesteps.
>
> To clarify this point, we have now added a new paragraph in the introduction with additional references.
>
> *Q2. Lack of Definitions and References In Figure 1, the term "ohe(type)" is used without definition or context, making it unclear to the reader.*
>
> **Response**: "ohe" refers to one-hot encoding, which is a standard abbreviation used in the field. We have now clarified this in the figure caption.
>
> *Q3. The "squareplus" activation function is mentioned but not cited.*
>
> **Response**: We have now cited the squareplus activation function.
> Barron, J.T., 2021. Squareplus: A softplus-like algebraic rectifier. arXiv preprint arXiv:2112.11687.
>
> *Q4. Limited Benchmarking: The paper restricts its experiments to simple spring systems, providing only a narrow validation of its methodology. Prior work in this domain typically includes experiments on 3-4 different datasets to establish the method's applicability.*
>
> **Response:** To the best of the authors' knowledge, this is the first attempt to learn Brownian dynamics directly from the trajectory in a physics-informed fashion. Note that, in contrast to deterministic systems, Brownian systems, are significantly more challenging to simulate. Being one of the first works, we have focused on fairly simple systems, namely, linear and non-linearly spring and binary spring systems. However, we have very critically analysed the performance of the models with several new baselines including MIBDGNN that has been now added to the manuscript. We aim to extend this to more complex systems and experimental data on colloidal gels, for example, as part of future work. This is now included in the limitations and future work.
>
> *Q5. Conclusion: While the paper introduces a GNN-based method for simulating over-damped Langevin dynamics, it suffers from multiple critical flaws, including a lack of clear motivation, insufficient references, and limited benchmarking. These drawbacks severely compromise the paper's value and applicability. Therefore, I recommend rejecting this submission.*
>
> **Response**: We have now added 13 additional references, a new paragraph on related work that further demonstrates the motivation, and added a new baseline model MIBDGNN that is evaluated on all the datasets. I hope the reviewer now finds the manuscript acceptable for publication. If there are any additional concerns, we request the reviewer to raise those.
>
> *Q6. Questions: How hard is it to simulate Brownian motions? If not hard, w.r.t. simulation time or numerical or modeling challenges, there is no point in replacing it with NN? If otherwise, please show the challenges in your draft.*
>
> **Response**: As outlined earlier, the main goal of the work is to learn the dynamics of Brownian systems directly from the trajectory. This is a challenging problem and more difficult than deterministic systems due to the inherent noise in the data. We demonstrate one of the first approaches to learn Brownian dynamics directly from the trajectory data of the systems. Traditional approaches relies on trial and error method to match the trajectory and may still be limited in terms of the functional forms of interactions. Replacing this traditional approach with neural network, enables the learning of any complex interactions directly from the trajectory data. This has now been clearly outlined in the manuscript.
>
> **Appeal to the reviewer:** With the additional experiments, results, and explanations, we now hope the reviewer finds the manuscript suitable for publication. Accordingly, we request you to raise the score for the manuscript. Please do let us know if there are any further queries.

---

> ### Author Response · Authors · 2023-11-20
> **Awaiting feedback!**
>
> Dear Reviewer Va4f,
>
> We thank you for taking the time to provide constructive comments, which have significantly improved the quality of the manuscript. Since we are in the last two days of the author-reviewer discussion period, we hope to engage in a discussion and improve the paper to the best extent possible. Specifically, the major changes made in response to the comments by the Reviewer are outlined below.
>
> 1. Updated the definitions in Figure 1.
> 2. Included a paragraph on motivation and related works.
> 3. We have added several additional references, hyperparametric optimization.
> 4. Finally, we have now added a new baseline, namely, MIBDGNN. The results of this model are included for all datasets and systems.
>
> With these additional experiments and improved explanations, we hope we have addressed all the concerns raised by the reviewer. If there are any outstanding concerns, we request the reviewer to please raise those. Otherwise, we would really appreciate it if the reviewer could increase the score.
>
> Looking forward to your response.
>
> Thank you,
>
> Authors

---

> > ### Author Response · Authors · 2023-11-21
> > **Eager to get your feedback on our revision**
> >
> > Dear Reviewer,
> >
> > There is less than 24 hours left till the completion of the author-reviewer discussion phase. We are extremely eager to know if there are any outstanding concerns that needs to be addressed. If not, we would really appreciate if you could support our work by increasing its rating.
> >
> > regards,
> >
> > Authors

---

> > > ### Comment · Reviewer_Va4f · 2023-11-22
> > > **Reply to reply**
> > >
> > > Dear authors,
> > >
> > > After checking your revised version. I believe it's in much better shape hence would increase the score to 6.
> > >
> > > Best

---

> > > > ### Author Response · Authors · 2023-11-22
> > > > **Thank you!**
> > > >
> > > > We thank you for the positive feedback and for supporting the work.

---

### Official Review · Reviewer_cdjq · 2023-10-30

**Soundness:** 4 excellent
**Presentation:** 3 good
**Contribution:** 3 good
**Rating:** 8
**Confidence:** 4

**Summary:**

The paper proposed Brownian graph neural networks (BROGNET) which is a new framework combining stochastic differential equations and graph neural networks to learn Brownian motion dynamics directly from trajectories. The architecture ensures linear momentum conservation, leading to improved learning of dynamics. Several baselines were proposed for comparison due to the limited existing benchmarks. The BROGNET's distinctive momentum conservation feature made it significantly superior to all other baselines. It also demonstrated the ability to generalize to much larger system sizes and different temperatures than those seen during training.

**Strengths:**

1. I do like the design of predicting interacting forces rather than total forces on each node which naturally conserves total momentum conservation. Such “hard constraint” (or physics-based inductive biases in the paper) not only rigorously adheres to physical principles but also significantly enhances the model's performance, compared with more straight forward methods such as adding regularizers to penalize the physics violation. I suggest adding a few sentences in the first paragraph to explicit distinguish the “hard constraints” vs “soft constraints”.

There are a few additional papers in this trajectory worth mentioning. The natural constraint design in this paper is more less like ref [A], as the constraint is pure summation. While it can be generalized as special cases of Ref[B],[C],[D] as well.

[A] A machine learning-aided global diagnostic and comparative tool to assess effect of quarantine control in COVID-19 spread, Patterns, 2020.

[B] ConCerNet: A Contrastive Learning Based Framework for Automated Conservation Law Discovery and Trustworthy Dynamical System Prediction. ICML 2023.

[C] Learning Physical Models that Can Respect Conservation Laws. ICML 2023.

[D] Unravelling the performance of physics-informed graph neural networks for dynamical systems. NeurIPS, 2022. (this is already cited.)

2. The model can generalize to unfamiliar system sizes and temperatures with zero-shot learning.

3. The outperforms existing baselines in various tasks.

**Weaknesses:**

1.	My major concern lies on the comparison baselines. Firstly, the prior work baselines (BFGN, BNequIP) do not seem very strong to me. However, I’m not an expert in partical-based systems and I’m not expecting each accepted paper will include comparisons with all the popular models. More importantly, to show the benefits of the physics-informed inductive bias, it is worth comparing the other model with the same backbone structure + training with regularization on the physics violation. I appreciate the ablation study with BDGNN which essentially shares the same backbone model. But adding a comparison experiment by training BDGNN with regularizing on momentum conservation will better prove the power of inductive bias.

2.	Regarding the sde integrator, I’m a bit concerned about the gradient stability due to the stochastic integral. Did you meet any issues when the random distribution of noise term leading to large error during back-propagation? Or do you need some sampling method to get the approximation of the drift term along with the variance of the stochastic term?

**Questions:**

1.	Typos: 2nd line under equation 7: ativation

2.	Is there any specific reason to use squareplus as activation function? I understand the comparison experiments with ReLU in appendix H which shows similar performance, but why squareplus is chosen at firsthand as it’s less popular?

3.	Equation 10 is questionable. \Delta \Omega should have the magnitude of \Delta t, but the following sentence mentioned “is a random number sampled from a standard Normal distribution”. Do you miss a magnitude of \Delta t?

4.	What does || || mean in equation 5,6?

---

> ### Author Response · Authors · 2023-11-19
> **Response to Reviewer cdjq: part 1**
>
> *Q1. I do like the design of predicting interacting forces rather than total forces on each node which naturally conserves total momentum conservation. Such “hard constraint” (or physics-based inductive biases in the paper) not only rigorously adheres to physical principles but also significantly enhances the model's performance, compared with more straight forward methods such as adding regularizers to penalize the physics violation. I suggest adding a few sentences in the first paragraph to explicit distinguish the “hard constraints” vs “soft constraints”.*
>
> **Response**: We have now added new text in the introduction. Moreover, we have now added a new model where "soft constraints" are used, namely, MIBDGNN, in addition to BroGNet.
>
> *Q2. There are a few additional papers in this trajectory worth mentioning. The natural constraint design in this paper is more less like ref [A], as the constraint is pure summation. While it can be generalized as special cases of Ref[B],[C],[D] as well.
> [A] A machine learning-aided global diagnostic and comparative tool to assess effect of quarantine control in COVID-19 spread, Patterns, 2020.
> [B] ConCerNet: A Contrastive Learning Based Framework for Automated Conservation Law Discovery and Trustworthy Dynamical System Prediction. ICML 2023.
> [C] Learning Physical Models that Can Respect Conservation Laws. ICML 2023.
> [D] Unravelling the performance of physics-informed graph neural networks for dynamical systems. NeurIPS, 2022. (This is already cited.)*
>
> **Response:** Thank you for pointing out this literature. We have now cited and discussed these papers in the introduction, first paragraph.
>
>
> *Q3. My major concern lies on the comparison baselines. Firstly, the prior work baselines (BFGN, BNequIP) do not seem very strong to me. However, I’m not an expert in particle-based systems and I’m not expecting each accepted paper will include comparisons with all the popular models. More importantly, to show the benefits of the physics-informed inductive bias, it is worth comparing the other model with the same backbone structure + training with regularization on the physics violation. I appreciate the ablation study with BDGNN which essentially shares the same backbone model. But adding a comparison experiment by training BDGNN with regularizing on momentum conservation will better prove the power of inductive bias.*
>
> **Response:** We thank the reviewer for the valuable insight. To test the hypothesis, we have now added a new baseline, namely, momentum-informed BDGNN (MIBDGNN). Interestingly, we observe that MIBDGNN occasionally outperforms BroGNet in predicting the dynamics. However, BroGNet outperforms MIBDGNN in momentum conservation as in the former case momentum conservation is enforced as a hard constraint, whereas in the latter case, it is a soft constraint. We have now included MIBDGNN for all the systems and included it in all the plots and results and discussion.
>
> We believe this has now significantly improved the quality of the manuscript. We thank the reviewer for the critical thinking and invaluable feedback.
>
> *Q4. Regarding the sde integrator, I’m a bit concerned about the gradient stability due to the stochastic integral. Did you meet any issues when the random distribution of noise term leading to large error during back-propagation? Or do you need some sampling method to get the approximation of the drift term along with the variance of the stochastic term?*
>
> **Response:** We thank the reviewer for raising this point. Indeed, we faced issues while training the model. However, effective hyperparametric optimization of the weights associated with the first and the second terms of the loss functions enabled stable training.
>
> *Q5. Typos: 2nd line under equation 7: activation*
>
> **Response:** Thank you for pointing it out. We have corrected them in the revision.

---

> > ### Author Response · Authors · 2023-11-19
> > **part 2**
> >
> > *Q6. Is there any specific reason to use squareplus as activation function? I understand the comparison experiments with ReLU in appendix H which shows similar performance, but why squareplus is chosen at firsthand as it’s less popular?*
> >
> > **Response**: We used SquarePlus in order to potentially extend BrogNet to learn energy instead of force in future work. In this case, the activation function needs to be double differentiation. In order to evaluate the role of activation function, we have performed extensive comparisons with SquarePlus, ReLU, LeakyReLU and Sigmoid in Appendix H. From the results, we observe that the results from different activation functions are comparable. To clarify, following text were added:
> > > Here, we evaluate the performance of BrogNet with respect to different activation functions: SquarePlus, ReLU, LeakyReLU and Sigmoid. From Fig. K, we observe that all the activation functions give comparable performance. We used SquarePlus in order to potentially extend BrogNet to learn energy instead of force in future work. In this case, the activation function needs to be double differentiation.
> >
> > *Q7. Equation 10 is questionable. \(\Delta \Omega\) should have the magnitude of \(\Delta t\), but the following sentence mentioned “is a random number sampled from a standard Normal distribution”. Do you miss the magnitude of \(\Delta t\)?*
> >
> > **Response:** We used the standard notation used in Ito calculus for writing stochastic differential equation (Euler-Maruyama integrator), namely, $dY_t = \theta (\mu-Y_t)dt + \sigma dW_t$, where $W_t$ represents the Wiener process, [1].
> >
> > [1] Kloeden, P.E. & Platen, E. (1992). Numerical Solution of Stochastic Differential Equations. Springer, Berlin
> >
> > *Q8. What does $||$ $||$ mean in equation 5,6?*
> >
> > **Response:** $||$ is used to denote the vector concatenation operation. We have now explicitly clarified this in the manuscript.
> >
> > **Appeal to the reviewer:** With the additional experiments, results, and explanations, we now hope the reviewer finds the manuscript suitable for publication. Accordingly, we request you to raise the score for the manuscript. Please do let us know if there are any further queries.

---

> ### Comment · Reviewer_cdjq · 2023-11-19
> **review reply**
>
> I thank the authors for the rebuttal.
>
> The added MIBDGNN experiment makes the comparison much stronger, at least from a research methodology perspective. This step is critical to prove the "hard constraints" outperform the regularizers in terms of conserving the quantities. And I appreciate the authors being candid with potentially better performance in the coordinates/KL divergence error with the soft constraints. This is expected because the structure's hard constraints with less model capacity can compromise the optimization. In fact, when people from specific backgrounds (particle simulation) use these ml-based models, they care more about physics constraint metrics than the coordinates/KL divergence error, because the former serves as a first-hand check of the credibility of the model for these users.
>
> The four papers I suggested can be categorized into "NN structures enforcing linear constraints of output", which is exactly what this paper aims to do (summation of total force equals 0). Are you citing them in the updated appendix? I didn't see it in the main paper though. I feel like it's even worth opening a small paragraph to discuss this branch of methods as this paper also falls into.
>
> My other questions are well answered. I checked the updated manuscript and can see the quality improvement. I will raise my score to 8.

---

> > ### Author Response · Authors · 2023-11-20
> > **Thank you and follow-up**
> >
> > We thank the reviewer for the positive feedback and for raising the score. We have modified the PDF and included the citations in the manuscript now with a few associated text. We, once again, thank the reviewer for the insightful feedback and appreciating the work for what it is.

---

### Official Review · Reviewer_VaBU · 2023-10-31

**Soundness:** 3 good
**Presentation:** 3 good
**Contribution:** 3 good
**Rating:** 6
**Confidence:** 4

**Summary:**

This paper introduces BroGNet, a GNN that is momentum conservative designed based on SDEs for Brownian dynamics learning.
The authors provide a very thorough introduction and related work section to motivate the paper and to provide the reader with sufficient background. Then, the method is presented, followed by several experiments in different scenarios. The proposed method seems to significantly outperforms existing methods.

**Strengths:**

The paper is mostly easy to follow and read.

The authors provide a very good background section to explain different terms, such that even non expert readers can understand the paper.

The experimental section looks promising under various settings.

The authors provide good explanations about the baselines and the experiment details.

**Weaknesses:**

Missing literature about GNNs: while this paper is concerned with learning brownian dynamics from data, there is a complementary topic in GNNs and that is the design of GNN architectures inspired by ODEs. I believe that the authors should add a discussion to the related work section to clarify the difference between the two. Some references are provided in [1-5].

Missing literature about Neuro ODEs: please see [6,7].

It is not clear why the authors propose to use the square plus activation. Is there a specific reason? (besides the experimental result provided in the appendix)

Reading the appendix, I understand that the authors used only one message passing layer in their implementation. Can you please elaborate on this point? What would the performance be like when adding more layers?

Refereces:

[1] Graph Neural Ordinary Differential Equations

[2] GRAND: Graph Neural Diffusion

[3] PDE-GCN: Novel Architectures for Graph Neural Networks Motivated by Partial Differential Equations

[4] Anti-Symmetric DGN: a stable architecture for Deep Graph Networks

[5] Graph-Coupled Oscillator Networks

[6] Stable Architectures for Deep Neural Networks

[7] Deep learning-based numerical methods for high-dimensional parabolic partial differential equations and backward stochastic differential equations

**Questions:**

Regarding the dynamic graph used here, how different is the proposed procedure than [8] ?

Regarding equation (8), this seems a bit like a discretized version of an advection operator (see [9,10]) for example. Can the authors expand on this point and clarify the differences?

[8] Dynamic Graph CNN for Learning on Point Clouds

[9] ADR-GNN: Advection-Diffusion-Reaction Graph Neural Networks

[10] Advective Diffusion Transformers for Topological Generalization in Graph Learning

---

> ### Author Response · Authors · 2023-11-19
> **Response to Reviewer VaBU**
>
> *Q1. Missing literature about GNNs: while this paper is concerned with learning brownian dynamics from data, there is a complementary topic in GNNs and that is the design of GNN architectures inspired by ODEs. I believe that the authors should add a discussion to the related work section to clarify the difference between the two. Some references are provided in [1-5].*
>
> **Response:** Thank you for raising this point. We have now added an additional paragraph in the introduction related to GNNs for modeling physical systems and included several additional references.
>
> *Q2. Missing literature about Neuro ODEs: please see[6,7]*
>
> **Response:** We have now added a new paragraph in the introduction and briefly discussed graph neural ODEs, and other such physics-informed GNNs. Additional references are included.
>
> *Q3. It is not clear why the authors propose to use the square plus activation. Is there a specific reason? (besides the experimental result provided in the appendix)*
>
> **Response:** In order to evaluate the role of activation function, we have performed extensive comparisons with SquarePlus, ReLU, LeakyReLU and Sigmoid in Appendix H. From the results, we observe that the results from different activation functions are comparable. We used SquarePlus in order to potentially extend BrogNet to learn energy instead of force in future work. In this case, the activation function needs to be double differentiation. To clarify, following text were added:
> > Here, we evaluate the performance of BrogNet with respect to different activation functions: SquarePlus, ReLU, LeakyReLU and Sigmoid. From Fig. K, we observe that all the activation functions give comparable performance. We used SquarePlus in order to potentially extend BrogNet to learn energy instead of force in future work. In this case, the activation function needs to be double differentiation.
>
>
> *Q4. Reading the appendix, I understand that the authors used only one message passing layer in their implementation. Can you please elaborate on this point? What would the performance be like when adding more layers?*
>
> **Response:** We have now added extensive hyperparametric optimization and included this as Appendix K 'Performance w.r.t. hyperparameters'. Moreover, the following text is added to the Appendix K.
>
> > PERFORMANCE W.R.T. HYPERPARAMETERS
> Here, we evaluate the performance of BroGNet w.r.t. number of layers of message passing [1, 2, 3] and the number of hidden layer neurons [5, 8, 16, 32]. In the paper for BroGNet, we choose the number of layers of message passing = 1 and the number of hidden layer neurons = 5. For other architectural details, check Appendix E
>
>
> *Q5. Regarding the dynamic graph used here, how different is the proposed procedure than [8]?*
>
> **Response:** Note that the approach presented in Ref. [8] is one of the earlier graph representations to represent any point cloud using a node and edge embedding. The present work focusses on learning the dynamics of Brownian systems. In this case, the GNN is used along with the physics-based equations to learn the dynamics. Note that different graph architectures can be used for the GNN. In the present work, we show the effect of different SOTA architectures including full graph network (BFGN), equivariant GNNs (BNequIP), and our own graph architecture, BroGNet. One of the major novelty of our architecture is its inherent ability to conserve momentum due to the directional edges.
>
> An additional paragraph is now added in the introduction on GNNs and physics-informed GNNS.
>
> *Q6. Regarding equation (8), this seems a bit like a discretized version of an advection operator (see [9,10]). Can the authors expand on this point and clarify the differences?*
>
> **Response:** We believe the reviewer is referring to Eq.(2) and Eq.(10). Indeed, Eq.(2) refers to a generic Brownian dynamics equations and Eq.(10) represents the numerical integration of this equation using Euler-Maruyama integrator. This is similar to Eq.(1) of the reference [9], which also represents the stochastic differential equation. We have added all the references below to the main manuscript.
>
> **References:**
> [1] Graph Neural Ordinary Differential Equations
> [2] GRAND: Graph Neural Diffusion
> [3] PDE-GCN: Novel Architectures for Graph Neural Networks Motivated by Partial Differential Equations
> [4] Anti-Symmetric DGN: a stable architecture for Deep Graph Networks
> [5] Graph-Coupled Oscillator Networks
> [6] Stable Architectures for Deep Neural Networks
> [7] Deep learning-based numerical methods for high-dimensional parabolic partial differential equations and backward stochastic differential equations
> [8] Dynamic Graph CNN for Learning on Point Clouds
> [9] ADR-GNN: Advection-Diffusion-Reaction Graph Neural Networks
> [10] Advective Diffusion Transformers for Topological Generalization in Graph Learning

---

> > ### Comment · Reviewer_VaBU · 2023-11-19
> > **Reviewer response**
> >
> > I thank the reviewers for the detailed response to my review and to other reviews. I am happy to continue supporting my score of weak acceptance and raise the confidence from 3 to 4, given the clarifications the authors made.

---

> > > ### Author Response · Authors · 2023-11-20
> > > **Thank you!**
> > >
> > > Thank you for the positive feedback and supporting the work.

---

### Official Review · Reviewer_KKvb · 2023-11-06

**Soundness:** 3 good
**Presentation:** 3 good
**Contribution:** 3 good
**Rating:** 6
**Confidence:** 4

**Summary:**

In this paper, the authors introduce an innovative framework, Brownian graph neural networks (BROGNET), that integrates stochastic differential equations and GNNs to directly learn Brownian dynamics from trajectories. Their method enforces the conservation of linear momentum within the system, leading to empirically observed improved performance in learning dynamics. The authors showcase the effectiveness of BROGNET by applying it to various benchmarked Brownian systems. They also demonstrate its ability to generalize to simulate previously unseen system sizes and temperatures

**Strengths:**

The main idea of the paper is interesting. It is well-written and the proposed method seems to be novel.

**Weaknesses:**

Some parts are unclear and require further explanations. There are questions and vague points that need addressing:

1. How does the suggested framework manage noisy or incomplete trajectory data, and is it capable of accurately learning the underlying  dynamics in such cases?

2. How does the choice of activation function affect the performance of the MLPs? Were other activation functions considered, and if so, how did they compare to the chosen function?

3. Can the MLP be replaced with other types of neural networks, such as convolutional neural networks or recurrent neural networks? How would this affect the performance of the proposed framework?

4. Can you provide more details on the scalability of the proposed framework? How does the computational complexity scale with the number of particles, and how does this affect its applicability to large-scale systems?

5. How were the hyperparameters chosen, and how does the choice of hyperparameters affect the performance of the proposed framework?

6. How the choice of graph topology affects the performance of the proposed framework? Were other graph topologies considered, and if so, how did they compare to the chosen topology?

7. How does the proposed method handle systems with external fields or other sources of non-deterministic forces?

8. Can you provide more details on the benchmarked Brownian systems used to evaluate BROGNET's performance? How do these systems compare to real-world applications?

I am happy to increase my score if the authors could address my concerns.

**Questions:**

Please see above!

---

> ### Author Response · Authors · 2023-11-19
> **Response to Reviewer KKvb**
>
> We thank the reviewer for the positive comments. Please find the point-by-point response below to each of the comments.
>
> *Q1. How does the suggested framework manage noisy or incomplete trajectory data, and is it capable of accurately learning the underlying dynamics in such cases?*
>
> **Response:** We thank the reviewer for raising this relevant point. In this work, we are learning Brownian dynamics which inherently consists of a drift (deterministic) term governed by interparticle forces and diffusion (stochastic) term governed a white noise. We draw the attention to the following points.
> 1. Learning from noisy data: In BrogNet, we are learning the Brownian dynamics from noisy data. Specifically, the ground truth consists of a Gaussian noise defined by 0 mean and a given standard deviation. We show that the BrogNet, after training, learns the standard deviation
> 2. Learning from one step: BrogNet employs a physics-informed approach where the input is the state of the system at time $t$ and the output is the state of the system at time $t + \Delta t$. This dynamics is learned in a statistical fashion using the Gaussian negative loglikelihood loss function. Thus, BrogNet only needs pair of steps and not a full trajectory.
>
> Altogether, BrogNet can learn from noisy and incomplete trajectory data.
>
> In order to demonstrate this further, we performed an additional experiment evaluating all the models with different standard deviations (see App. I, ‘Generalizability to unseen $\gamma$’). The following new text is added to the App. I along with a new Fig. N.
> > Here, we evaluate BroGNet performance with respect to different $\gamma$ values: 0.1, 0.6, 0.8, 1.0, 1.2, 1.4, and 10. As $\gamma$ increases performance improves. This is due to the fact that for a higher value of $\gamma$, there will be a smaller standard deviation for the stochastic term.
>
> *Q2. How does the choice of activation function affect the performance of the MLPs? Were other activation functions considered, and if so, how did they compare to the chosen function?*
>
> **Response:** In order to evaluate the role of activation function, we have performed extensive comparisons with SquarePlus, ReLU, LeakyReLU and Sigmoid in Appendix H. From the results, we observe that the results from different activation functions are comparable. We used SquarePlus in order to potentially extend BrogNet to learn energy instead of force in future work. In this case, the activation function needs to be double differentiation. To clarify, following text were added:
> > Here, we evaluate the performance of BrogNet with respect to different activation functions: SquarePlus, ReLU, LeakyReLU and Sigmoid. From Fig. K, we observe that all the activation functions give comparable performance. We used SquarePlus in order to potentially extend BrogNet to learn energy instead of force in future work. In this case, the activation function needs to be double differentiation.
>
> *Q3. Can the MLP be replaced with other types of neural networks, such as convolutional neural networks or recurrent neural networks? How would this affect the performance of the proposed framework?*
>
> **Response:** We thank the reviewer for this comment. In the present work, we use a graph neural network to model the physical system. This is important as we need to capture the topology of the system. Moreover, the baselines chosen in the work, namely, NN and BNN uses an MLP instead of a GNN. We observe that these models give poor performance in comparison to BrogNet and other GNN versions. This confirms that the topology of the structure is important. Moreover, another important property that GNNs exhibit is permutation invariance. CNNs and RNNs do not exhibit this property and hence cannot be effectively used in the present scenario.
>
> *Q4. Can you provide more details on the scalability of the proposed framework? How does the computational complexity scale with the number of particles, and how does this affect its applicability to large-scale systems?*
>
> **Response:** Scalability is an important aspect in modeling such systems as realistic systems may have a large number of degrees of freedom. To demonstrate this, we show the performance of models trained on 5-particle systems on 50, 500, and 5000 particle systems. To show the computational time with the number of particles a Fig. K and following text were added in Appendix F:
>
> > Further, Fig. K shows the Inference time w.r.t number of particles for BFGN, BDGNN, MIBDGNN, BROGNET and BNEQUIP

---

> > ### Author Response · Authors · 2023-11-19
> > **part 2**
> >
> > *Q5. How were the hyperparameters chosen, and how does the choice of hyperparameters affect the performance of the proposed framework?*
> >
> > **Response:** Hyperparameters are chosen based on grid-search. To clarify this, we have added Appendix K 'Performance w.r.t. hyperparameters'. Moreover, following text is added:
> >
> > > PERFORMANCE W.R.T. HYPERPARAMETERS
> > Here, we evaluate the performance of BroGNet w.r.t. number of layers of message passing [1, 2, 3] and the number of hidden layer neurons [5, 8, 16, 32]. In the paper for BroGNet, we choose the number of layers of message passing = 1 and the number of hidden layer neurons = 5. For other architectural details, check Appendix E
> >
> > *Q6. How the choice of graph topology affects the performance of the proposed framework? Were other graph topologies considered, and if so, how did they compare to the chosen topology?*
> >
> > **Response:** To evaluate the role of graph topology, we performed additional experiments where the topology is connected randomly. We observe that the performance decreases significantly when graph topology is random.
> >
> > To clarify this, we have added a new Appendix J. ‘Importance of Graph Topology', where we evaluate the role of graph topology. Further, following new text is added:
> > > Here, we evaluate BroGNeT when there is correct graph topology vs BroGNeTRND when there is random connection between senders and receivers. From Fig. M it is clear that performance is significantly high with correct graph topology
> >
> > *Q7. How does the proposed method handle systems with external fields or other sources of non-deterministic forces?*
> >
> > **Response:** We thank the reviewer for raising this interesting point. At present, BroGNet cannot handle external fields. This could be achieved by an architectural modification where an additional learnable nodal force could be added as an MLP in the architecture. This is now included in the limitations and future work.
> >
> > *Q8. Can you provide more details on the benchmarked Brownian systems used to evaluate BROGNET's performance? How do these systems compare to real-world applications?*
> >
> > **Response:** To the best of the authors' knowledge, this is the first framework that proposes to learn Brownian dynamics directly from the data. This can be used to study several real-world problems such as biological systems, colloidal gels, proteins, and even planetary systems. The model systems of harmonic and non-linear springs are the simplest systems that represent Brownian dynamics. However, it should be noted that even proteins and other biological molecules are also modeled as Harmonic springs. Thus, the present work can be extended to more realistic datasets and experimental datasets as part of future work. This is now included in the Limitations and future works section.
> >
> > **Appeal to the reviewer:** With the additional experiments, results, and explanations, we now hope the reviewer finds the manuscript suitable for publication. Accordingly, we request you to raise the score for the manuscript. Please do let us know if there are any further queries.

---

> > > ### Comment · Reviewer_KKvb · 2023-11-21
> > > **Thank you!**
> > >
> > > I greatly appreciate the authors for their detailed response and clarifications! My concerns have been addressed, and I will increase my score to 6.

---

> ### Author Response · Authors · 2023-11-20
> **Awaiting feedback!**
>
> Dear Reviewer Va4f,
>
> We thank you for taking the time to provide constructive comments, which have significantly improved the quality of the manuscript. Since we are in the last two days of the author-reviewer discussion period, we hope to engage in a discussion and improve the paper to the best extent possible. Specifically, the major changes made in response to the comments by the Reviewer are outlined below.
>
> 1. Generalizability to **unseen $\gamma$**.
> 2. Additional evaluation on the **effect of activation functions**.
> 3. Additional studies on **hyperparametric optimization**.
> 4. **Scalability** (in terms of inference time) to system sizes **three orders of magnitude** larger than the training system.
> 5. Additional experiments on **graph topology**.
>
> Finally, we have now added **a new baseline, namely, MIBDGNN**. The results of this model are included for all datasets and systems.
>
> With these additional experiments and improved explanations, we hope we have addressed all the concerns raised by the reviewer. If there are any outstanding concerns, we request the reviewer to please raise those. Otherwise, we would really appreciate it if the reviewer could increase the score.
>
> Looking forward to your response.
>
> Thank you,
>
> Authors

---

> > ### Author Response · Authors · 2023-11-21
> > **Eagerly awaiting feedback from Reviewer KKvb**
> >
> > Dear Reviewer KKvb,
> >
> > We thank you for the insightful comments on our work. Your suggestions have now been incorporated in our revision and we are eagerly waiting for your feedback.
> >
> > We are particularly encouraged by your comment
> >
> > > "I am happy to increase my score if the authors could address my concerns."
> >
> > As the author-reviewer discussion phase is approaching its conclusion in just a few hours, we are reaching out to inquire if there are any remaining concerns or points that require clarification. Your support in this final phase, particularly if you find the revisions satisfactory, would be immensely appreciated.
> >
> > regards,
> >
> > Authors

---

### Author Response · Authors · 2023-11-19
**Summary of the rebuttal**

We thank the reviewers for the careful evaluation and suggestions. Please find a point-by-point response to all the comments raised by the reviewers below. We have also updated the main manuscript and the appendix to address these comments. The changes made in the main manuscript are highlighted in *blue* color. The major changes made in the manuscript are listed below.
1. **Momentum-informed BDGNN (MIBDGNN)**: We have now included a new mode, namely, MIBDGNN, where the momentum conservation is enforced as a soft-constrained instead of a hard constraint as in BroGNet. Interestingly, we observe that MIBDGNN occasionally outperforms BroGNet in predicting the dynamics, whereas, BroGNet exhibits superior momentum conservation. MIBDGNN is evaluated on all the systems and datasets and is included in all the plots and tables of the manuscript.
2. **Generalizability to unseen standard deviation**: In order to evaluate the ability of the models to generalize to unseen $\gamma$ values, additional experiments are performed on seven different values of $\gamma$ (see **App. I, Fig. N** in the Appendix).
3. **Activation function**: In order to understand the effect of activation functions, we have now evaluated the models with four different activation functions (see **App. H, Fig. L**).
4. **Scalability to large systems:** We demonstrate the ability of BroGNet to scale systems three orders of magnitude larger than the training system (5 particle to 5000 particles). See **App. F, Fig. H**
5. **Hyperparametric optimization:** Extensive hyperparametric optimization has been performed using grid search and the results are included in **App. E, App. H, App. K**.
6. **Graph topology**: To evaluate the role of graph topology, we perform comparison of BroGNet with a BroGNet with random graph topology (see App. J).
7. **Related works on GNNs**: A paragraph is added in the introduction on related works using GNNs for modeling physical systems with 10 additional references.

---

### Meta-Review · Area_Chair_kJSH · 2023-12-10

**Metareview:**

This paper develops a graph neural network (GNN)-based approach to learn the dynamics of a stochastic multi-body system driven by Brownian motion. Each node of the graph represents one of the particles, e.g., one spring in an inter-connected system of springs and the GNN learns the drift and the (scalar) diffusion terms. The authors claim that since Brownian motion introduces zero force on average, the momentum of the learned system is conserved; their architecture is designed to capture this. Simulation experiments on toy examples consisting of linear and nonlinear springs are used to study this network.

All reviewers leaned towards accepting the paper and the authors have addressed and clarified a number of  implementation details in the updated manuscript. For the revised manuscript, it will be useful to motivate this problem better: for Brownian dynamics model to be valid, a physical system needs to have 100s if not 1000s of particles, and one may not have observations of the locations of all particles in such problems. For example, the authors give the example of colloidal systems or molecular dynamics in the introduction. It would be useful to show results on more realistic problems. The simulation experiments presented in this paper are far from any real problem.

**Justification For Why Not Higher Score:**

Please see above.

**Justification For Why Not Lower Score:**

The reviewers are in consensus that this paper meets the bar for acceptance.

---

### Decision · Program_Chairs · 2024-01-16

Accept (poster)